# Triggered contraction of self-assembled micron-scale DNA nanotube rings

Maja Illig [1,2,8], Kevin Jahnke [2,3,8], Lukas P. Weise [4,8], Marlene Scheffold [2], Ulrike Mersdorf[2], Hauke Drechsler [5,6], Yixin Zhang [5], Stefan Diez [5,7] ✉, Jan Kierfeld [4] ✉ & Kerstin Göpfrich [1,2] ✉

Contractile rings are formed from cytoskeletal filaments during cell division. Ring formation is induced by specific crosslinkers, while contraction is typically associated with motor protein activity. Here, we engineer DNA nanotubes and peptide-functionalized starPEG constructs as synthetic crosslinkers to mimic this process. The crosslinker induces bundling of ten to hundred DNA nanotubes into closed micron-scale rings in a one-pot self-assembly process yielding several thousand rings per microliter. Molecular dynamics simulations reproduce the detailed architectural properties of the DNA rings observed in electron microscopy. Theory and simulations predict DNA ring contraction – without motor proteins – providing mechanistic insights into the parameter space relevant for efficient nanotube sliding. In agreement between simulation and experiment, we obtain ring contraction to less than half of the initial ring diameter. DNA-based contractile rings hold promise for an artificial division machinery or contractile muscle-like materials.

Cell division is a hallmark of life. After duplication and separation of the genetic information, the cellular compartment has to be divided to give rise to two daughter cells. Nature's solution for compartment division is the formation of contractile rings made from cytoskeletal filaments[1]. In eukaryotic cells, micron-scale actomyosin rings assemble at the cell's equatorial plane at the end of mitosis and meiosis[2]. The ring contraction can be powered by two distinct mechanisms, namely molecular motor activity[3] and diffusable actin crosslinkers[4]. Whereas the first is an active, energy-consuming process, the latter mechanism mediates passive entropy driven contraction. Recently, it has been shown that passive filament crosslinkers can generate filament sliding and contractile forces, that are sufficient to antagonize motor protein action in microtubule[5] and actin networks[4].

Bottom-up synthetic biology has pursued the long-term goal to reconstitute a minimal division machinery inside lipid vesicles to establish fundamental physical principles in isolation of the complex environment of a cell and to eventually engineer a self-replicating cellular system from scratch. Towards this goal, actomyosin rings have been formed in vitro[6] and their contraction has been demonstrated in confinement[7]. This led to membrane deformations in lipid vesicles[8]. Nevertheless, the reconstitution of a division machinery that can complete the division of lipid vesicles remains an open challenge in bottom-up synthetic biology[9,10].

It is important to critically ask which physical features are required for a minimal division system and how they contribute to the contraction process. Towards this end, it would be ground breaking to establish an entirely synthetic division machinery, which does not rely on nature's building blocks.

A fully engineered contractile ring could yield mechanistic insights into the biophysics of the process towards an alternative set of

[1]Center for Molecular Biology of Heidelberg University (ZMBH), Heidelberg University, Im Neuenheimer Feld 329, 69120 Heidelberg, Germany. [2]Max Planck Institute for Medical Research, Biophysical Engineering Group, Jahnstraße 29, 69120 Heidelberg, Germany. [3]Harvard University, School of Engineering and Applied Sciences (SEAS), 9 Oxford Street, 02138 Cambridge, MA, USA. [4]TU Dortmund University, Department of Physics, Otto-Hahn-Str. 4, 44221 Dortmund, Germany. [5]B CUBE - Center for Molecular Bioengineering and Cluster of Excellence Physics of Life, Technische Universität Dresden, Tatzberg 41, 01307 Dresden, Germany. [6]Tübingen University, Center for Plant Molecular Biology (ZMBP), Auf der Morgenstelle 32, 72076 Tübingen, Germany. [7]Max Planck Institute of Molecular Cell Biology and Genetics, Pfotenhauerstrasse 108, 01307 Dresden, Germany. [8]These authors contributed equally: Maja Illig, Kevin Jahnke, Lukas P. Weise. ✉e-mail: stefan.diez@tu-dresden.de; jan.kierfeld@tu-dortmund.de; k.goepfrich@zmbh.uni-heidelberg.de

molecular components for the division of synthetic cells. There are ongoing efforts in the field of DNA nanotechnology to recreate functional mimics of cytoskeletal elements from DNA. Of particular interest are DNA nanotubes[11], which have been equipped with different features that mimic functions of a cytoskeleton, such as reversible assembly[12,13], directional growth[14], signaling[15], and transport with engineered molecular motors[16] or enzyme activity[17]. However, a key functionality remains unachieved, namely the formation of contractile DNA rings. Closed DNA rings have been assembled on the nanoscale[18,19]. They have been used to template liposomes[20] or gold nanoparticles[21], to engineer liposome fusion and lipid transfer[22] or mechanically interlocked molecules[23] and they have been used as large-diameter membrane pores[24]. However, these nanoscale DNA rings are one to two orders of magnitude too small to span the circumference of synthetic cellular compartments and beyond that, no mechanism for their contraction has been proposed or experimentally realized.

Therefore, our aim is the self-assembly of DNA-based contractile rings on the micron scale. We reason that the assembly and contraction can take inspiration from the mechanisms at play for nature's cytoskeletons. While ring assembly clearly requires crosslinkers, contraction could either be achieved with suitable molecular motors or only by passive crosslinkers, as it has been discovered recently[4,25]. Assuming that the passive crosslinker approach is more straight forward to adapt to DNA nanotubes, we require a synthetic crosslinker for DNA nanotubes. Multivalent positively charged peptides have been shown to crosslink microtubules[26].

Here, we revert to the DNA nanotubes as an entirely synthetic system that is well established in the bottom-up synthetic biology community as an alternative route to the reconstitution of proteins. The reconstruction of a protein-like machinery from a different material is not only exciting in itself, but it may also provide a shortcut towards a truly self-replicating system since DNA replication requires fewer components than the replication of proteins. In the long term, one could thus envision a synthetic cell that operates outside of the central dogma of molecular biology.

## Results and discussion

We show that we can bundle DNA nanotubes and achieve the one-pot self-assembly of closed micron-scale DNA rings upon addition of such multivalent positively charged peptides. We control the DNA bundle thickness as well as the ring diameter. With theory and molecular dynamics simulations we gain mechanistic insights into the formation of DNA nanotube rings and the architecture of its contraction

mechanism. We translate the simulation parameters of interest into physical properties of our system and realize the predicted conditions experimentally. Thereby, we achieve the contraction of the DNA rings to less than 45 percent of their initial diameter. We relate this to the theory and adapt it to the particularities of the physical system so that we can reduce the entangled relationships of the experiment to quantitative parameters.

### Synthetic peptides as crosslinkers for DNA nanotubes

We first assemble DNA nanotubes from the well-established double-crossover DNA tile design, whereby each tile consists of five DNA oligomers (Fig. 1a, Supplementary Table 1)[11]. Due to their sticky-end overhangs and their intrinsic curvature, these tiles assemble into hollow nanotubes consisting of 8 to 20 DNA duplexes (4 to 10 tiles)[11], resulting in an experimentally determined diameter of $11.8 \pm 2.1$ nm. To form bundles and contractile rings from these DNA nanotubes, we need a synthetic crosslinker that satisfies the following two requirements imposed by the nature of the DNA nanotubes: First, a crosslinker that binds to DNA nanotubes by electrostatic interactions has to be positively charged because of the negatively charged backbone of the DNA. Second, it needs to act as a multivalent crosslinker that can connect multiple DNA nanotubes. Thus, we can make use of a multivalent positively charged peptide construct, which has been shown to electrostatically crosslink microtubules[26].

The construct consists of a four-arm 10 kDa starPEG backbone which is coupled to seven lysine-alanine amino acid repeats on each of the four arms (Fig. 1b). We will refer to it as starPEG-(KA7)$_4$. The end-to-end distance $R_0$ of the polymer (i.e. a long chain with no significant hindrances to backbone rotation, as is the case for PEG) can be derived from treating the polymer as a self-avoiding freely jointed chain with $R_0 = b \cdot N^{(3/5)}$ with $b$, the Kuhn monomer (1.1 nm for PEG), and $N$, the number of Kuhn monomers in the chain (18.25 for 2.5 kDa PEG arm)[27]. According to this, each of the four arms has an apparent length of about 6.3 nm. By applying the formula $R_{min} = 0.066 M^{1/3}$ for $M$ in Da[28], one can estimate the minimal radius for alanine (89 Da) being 0.29 nm, and for lysine (146 Da) being 0.35 nm (PubChem release 2021.10.14). Thus, the peptide chain measures 4.5 nm. In a fully extended conformation, (for PEG treated as a freely jointed chain and the peptide as a linear chain) the starPEG construct would have a max. length of approximately 21.6 nm – long enough to compliantly connect two DNA nanotubes with a diameter of ~12 nm each. Hence, the construct exhibits four flexible arms with positively charged amino acid repeats that can bind to and crosslink the negatively charged DNA nanotubes

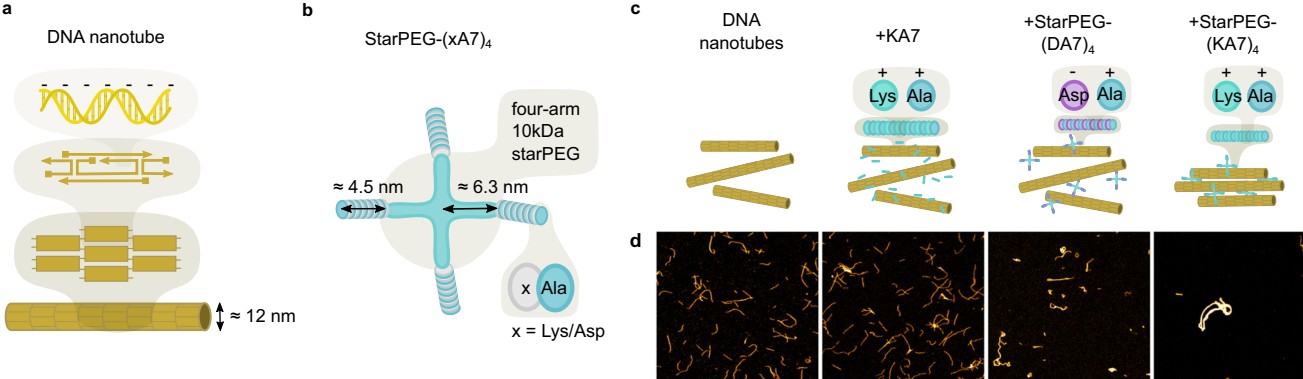

**Fig. 1 | StarPEG-(KA7)$_4$ bundles DNA nanotubes. a** Schematic illustration of DNA nanotubes formed from double-crossover DNA tiles[11]. **b** Schematic illustration of tetravalent starPEG-(xA7)$_4$ composed of four branches of 7 lysine- or aspartate-alanine repeats. **c** Schematic illustration of DNA nanotubes in the absence and presence of different synthetic peptide constructs. **d** Confocal images of DNA nanotubes (30 nM DNA tiles, labeled with Atto633, $\lambda_{ex} = 640$ nm) without any peptide; with 2 µM positively charged monovalent KA7-peptide; with 500 nM negatively charged tetravalent starPEG-(DA7)$_4$ composed of four branches of 7 aspartate-alanine repeats and with 500 nM positively charged tetravalent starPEG-(KA7)$_4$ composed of four branches of 7 lysine-alanine repeats (from left to right). Scale bar: 10 µm.

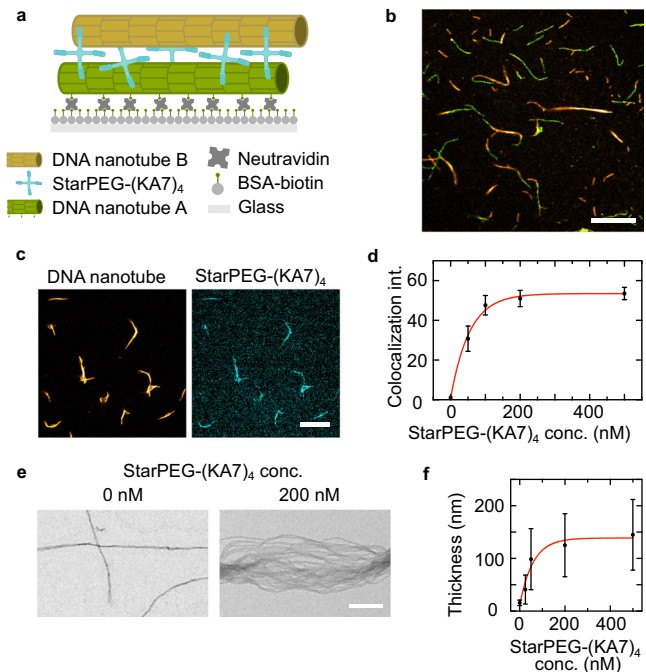

**Fig. 2 | DNA nanotube colocalization and bundling by starPEG-(KA7)₄.**
**a** Schematic illustration of the DNA nanotube colocalization assay with a bioti-nylated DNA nanotube A (green) bound to biotinylated bovine serum albumin-coated glass slides upon addition of 0.2 wt% neutravidin. StarPEG-(KA7)₄ crosslinks DNA nanotube B (yellow) to the immobilized nanotube A.
**b** Composite confocal image of the immobilized DNA nanotube A (green, 6-FAM, $\lambda_{ex}$ = 488 nm) and DNA nanotube B (yellow, Atto633-labeled, $\lambda_{ex}$ = 640 nm) illustrating their colocalization. Scale bar: 10 µm. **c** Confocal images of DNA nanotubes (30 nM DNA tiles, yellow, Atto633-labeled, $\lambda_{ex}$ = 640 nm) and 200 nM starPEG-(KA7)₄ (cyan, TAMRA-labeled, $\lambda_{ex}$ = 561 nm). Scale bar: 10 µm.
**d** Colocalization intensity of starPEG-(KA7)₄ (TAMRA-labeled, $\lambda_{ex}$ = 561 nm) with DNA nanotubes (30 nM DNA tiles) (Atto633-labeled, $\lambda_{ex}$ = 640 nm, Mean ± SD, $n$ = 10 DNA nanotube bundles analyzed per condition, exponential fit plotted as red line, $y = y_0 + (P - y_0)(1 - \exp(-kx))$, $y_0$ = 0.85, $P$ = 53.49, $k$ = 0.018). **e** TEM micrographs of DNA nanotubes (30 nM DNA tiles) in the absence of starPEG-(KA7)₄ (left) and in presence of 200 nM starPEG-(KA7)₄ (right). Scale bar: 200 nm. **f** DNA nanotube bundle thickness over starPEG-(KA7)₄ concentration (Mean ± SD, $n$ = 100 DNA nanotube bundles analyzed per condition, 30 nM DNA tiles, exponential fit plotted as red line, $y = y_0 + (P - y_0)(1 - \exp(-kx))$, $y_0$ = 11.79, $P$ = 138.80, $k$ = 0.017). Source data are provided as a Source Data file.

of bundling correlates with the concentration of starPEG-(KA7)₄ relative to the concentration of DNA tiles (Supplementary Fig. 1), forming a hybrid material with engineerable properties. The bundles curve into closed rings. To tune ring size and accomplish contraction, we first need to understand the action of the peptide crosslinker on DNA nanotubes. In the following sections, bundle formation is discussed in particular. The properties of ring formation and ring structures are then considered.

## Characterization of DNA nanotube crosslinking with synthetic multivalent peptides

In particular, we need to establish whether the DNA nanotube bundling is caused by the depletion effect only, whereby starPEG-(KA7)₄ acts as a molecular crowder, or whether the peptide can actually crosslink DNA nanotubes. We therefore immobilize DNA nanotubes on the surface of the observation chamber and pre-load them with starPEG-(KA7)₄. We then wash out excess starPEG-(KA7)₄ from the solution when adding a second type of DNA nanotubes, which we labeled with a different fluorophore in order to distinguish them from the immobilized DNA nanotubes (immobilized DNA nanotubes A: green; free DNA nanotubes B: yellow, Fig. 2a). After another washing step, only starPEG-(KA7)₄ that is bound to the DNA nanotube A is present and responsible for the colocalization of the two DNA nanotube types (Fig. 2b). Thereby we conclude that the binding must be induced by crosslinking and not by the depletion effect only.

To assess how much starPEG-(KA7)₄ can bind to DNA nanotubes, we label the synthetic starPEG peptides with fluorescent 5-TAMRA and analyze its colocalization with the DNA nanotubes (Fig. 2c), i.e. the fluorescence of starPEG-(KA7)₄ colocalizing with the DNA nanotubes (see 'Colocalization Assay' in the "Methods" section and Supplementary Fig. 2). The colocalization intensity saturates at around 200 nM starPEG-(KA7)₄ for DNA nanotubes assembled from 30 nM DNA tiles (Fig. 2d and Supplementary Fig. 3). The concentration at which binding saturates agrees with previous results for the microtubule binding of starPEG-(KA7)₄[26]. Additionally, we analyze the DNA bundle thickness with negative stain transmission electron microscopy (TEM) for varying starPEG-(KA7)₄ concentrations. Electron micrographs reveal the transition from single DNA nanotubes (single (11.8 ± 2.1) nm or weakly bundled (15.7 ± 5.3) nm) to DNA bundles consisting of tens of DNA nanotubes in the presence of starPEG-(KA7)₄ (Fig. 2e and Supplementary Fig. 4). Concomitantly, the bundle thickness increases by one order of magnitude from single DNA nanotubes with an apparent cross section of (15 ± 5) nm to bundles with a cross section of (145 ± 67) nm for 0 and 500 nM starPEG-(KA7)₄, respectively. In accordance with the colocalization intensity, the bundle thickness also does not increase further than at 200 nM starPEG-(KA7)₄ indicating a maximal occupancy of starPEG-(KA7)₄ on the DNA nanotubes (Fig. 2f). We expect both the association of starPEG-(KA7)₄ and the association of DNA nanotubes to bundles, to follow association binding kinetic models. Therefore, it can be fitted with an equation of the form: $y = y_0 + (P - y_0)(1 - \exp(-kx))$ with x being the starPEG-(KA7)₄ concentration. StarPEG-(KA7)₄ binding to DNA nanotubes saturates within minutes (Supplementary Fig. 5).

## Self-assembly of micron-scale DNA rings

To study the ability of starPEG-(KA7)₄ to promote the self-assembly of DNA nanotubes into higher-order structures, we observe DNA nanotubes without immobilization in an unconstrained 3D environment (Supplementary Movie 1). We find that starPEG-(KA7)₄ promotes the efficient formation of DNA nanotube rings with several micrometers in diameter. Figure 3a provides a confocal overview image that depicts the high abundance of ring-like DNA structures. The sample is taken

by electrostatic interactions. As a control, we additionally synthesize monovalent KA7 peptides, which are not expected to crosslink DNA nanotubes, as well as a construct that features seven negatively charged aspartate-alanine repeats on a tetravalent starPEG backbone (starPEG-(DA7)₄) which is not expected to bind to the negatively charged DNA backbone. All constructs are further labeled with 5-TAMRA ($\lambda_{ex}$ = 561 nm) to allow for their visualization with confocal microscopy.

To test whether starPEG-(KA7)₄ indeed can bundle DNA nanotubes, we mix fluorescently-labeled DNA nanotubes with starPEG-(KA7)₄ or the respective control constructs and image them by confocal microscopy (Fig. 1c, d). In the presence of monovalent KA7 peptides, single DNA nanotubes remain homogeneously distributed across the observation chamber similar to DNA nanotubes in absence of any peptide. Weak DNA nanotube bundling can be observed in the presence of the negatively charged starPEG-(DA7)₄ (Fig. 1d), likely due to the positively charged magnesium-ions in solution. Notably, in the presence of starPEG-(KA7)₄, the DNA nanotubes form DNA nanotube bundles with much higher fluorescence intensity extending to a length of several tens of micrometer (Fig. 1d, right image). The extent

directly from the storage solution without additional purification steps. We reproducibly obtain $5300 \pm 2500$ closed DNA rings per µL (mean ± SD, Supplementary Fig. 6). StarPEG-(KA7)$_4$ thus mimics the behavior of septins in actin filament networks[29]. The formation of closed rings and their microscopic structure are verified by STED (Fig. 3b and Supplementary Fig. 7) and transmission electron microscopy (Supplementary Fig. 7), which also reveals that the DNA rings typically consist of tens to hundred of DNA nanotubes. Since the persistence length to mean length ratio for DNA nanotubes (Poisson distributed lengths) is around 1 and the persistence length is defined as the length over which correlations in the direction of the tangent are lost, we can assume that an encountering of DNA nanotube ends is possible on the experimentally relevant time scale of several seconds up to a few minutes. Once the ends have met, some DNA nanotubes may undergo end-to-end joining, others overlap to maximize the crosslinking. STED reveals free DNA nanotube ends at the edges of the ring suggesting that the observed ring formation cannot be a result of end-to-end joining only, but is rather mediated by crosslinkers along the DNA nanotubes that induce further growth of the bundle thickness by recruiting more single or prebundled DNA nanotubes (Fig. 3b and Supplementary Fig. 7).

We analyze the ring diameter for a range of starPEG-(KA7)$_4$ to DNA tile ratios. For an excess of DNA tiles, the amount and the diameter of rings is significantly reduced compared to equimolar ratios (Fig. 3c, d). The ring diameter is, however, largely independent of the magnesium-ion concentration and the absolute concentrations of starPEG-(KA7)$_4$ or DNA tiles (Supplementary Fig. 8). The number of DNA rings in a given sample volume (Supplementary Figs. 5 and 6) as well as their diameter remains constant over time (Supplementary Fig. 9).

Taken together, we obtain the first self-assembled and free-standing micron-scale DNA nanotube bundle rings to the best of our knowledge. This ring formation mechanism seems to mimic naturally occurring ring formation by cytoskeletal filament crosslinking[4,29]. To gain a deeper understanding of DNA nanotube ring formation and to derive strategies for potential ring contraction, we next develop a theoretical framework and subsequently combine it with coarse-grained molecular dynamics (MD) simulations.

## Theory and predictions for DNA ring formation and contraction
Ring formation and contraction of bundles of semiflexible filaments such as DNA nanotubes can be described by a balance of an adhesion energy gain, which is reduced by a surface energy term and a bending energy term[30,31]. The adhesion energy gain can be, for example, crosslinker-mediated, electrostatic or from depletion attraction, whereas the surface energy term results from DNA nanotubes exposed to solvent and lacking adhesion. Crosslinkers generate an adhesive energy between two DNA nanotubes by presenting one adhesive end to each DNA nanotube[31]. Thereby they accumulate in the overlap region between DNA nanotubes and can be viewed as a one-dimensional gas of particles confined to the overlap region. We show that, generally, this dense and mobile gas of adhesive crosslinkers gives rise to an additional effective free energy of adhesion inside a nanotube bundle, which is of entropic nature. In case of actin filaments it has been confirmed that the entropy of the crosslinker gas tends to maximize overlaps between the filaments[4]. With our generalization, the same should hold true for DNA nanotubes.

We consider a torus of diameter $D$ consisting of bundled DNA nanotubes, which are uniformly bent (Fig. 4a). The bundle's circular cross section contains $N$ DNA nanotubes resulting in a total length $L_{\text{tot}} = \pi N D$. The bundle has a bending rigidity $\kappa_b(N)$, which is related to the bending rigidity $\kappa$ of individual DNA nanotubes via $\kappa_b(N) = \kappa N^\alpha$, with $\alpha = 1$ for decoupled sliding DNA nanotubes and $\alpha = 2$ if crosslinking resists shear[32]. We assume uniform DNA nanotube bending rigidities throughout the entire torus. We assume an adhesion energy $g$ per length for the DNA nanotubes in the interior of the bundle from crosslinker-mediated attraction or depletion attraction. For roughly circular cross sections, there should be $N_s = a_s N^{1/2}$ out of $N$ DNA nanotubes at the bundle surface with a geometric factor $a_s$ of order unity. This results in an effective DNA nanotube length $L_i = L_{\text{tot}} - \pi a_s N^{1/2} D$ within the interior of the DNA nanotube bundle that is fully accessible to the attraction of strength $g$.

Contraction of a toroidal bundle of DNA nanotubes can then be described by the free energy

$$F = E_{\text{bend}} + E_{\text{ad}} + F_c \qquad (1)$$

$$= 2\pi \kappa_b(N) D^{-1} - g L_i + F_c(L_i), \qquad (2)$$

which is the sum of bundle bending energy $E_{\text{bend}}$, adhesion energy in the interior $E_{\text{ad}}$ and the entropic free energy $F_c$ of the crosslinker gas in the interior of the bundle, which will arise if crosslinkers are mobile. We approximate $F_c$ by the entropic free energy of a Tonks gas of $N_c$ non-overlapping particles of crosslinker size $b_c$ in a one-dimensional

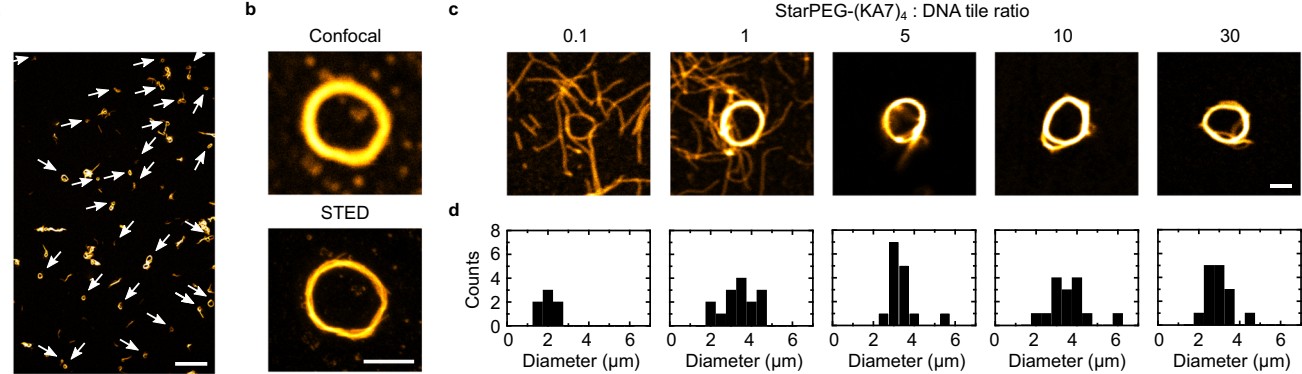

**Fig. 3 | StarPEG-(KA7)$_4$ induces self-assembly of multiple DNA nanotubes into closed micron-scale DNA rings. a** Confocal overview image of self-assembled DNA nanotube rings formed from 50 nM DNA tiles (yellow, Atto633-labeled, $\lambda_{ex} = 640$ nm) in presence of 500 nM starPEG-(KA7)$_4$. Rings are highlighted with white arrows. Scale bar: 20 µm. **b** Representative confocal (top) and STED (bottom) images of an individual DNA ring formed from 50 nM DNA tiles (yellow, Atto633-labeled, $\lambda_{ex} = 640$ nm) in presence of 500 nM starPEG-(KA7)$_4$. Scale bar: 2 µm.

**c** Representative confocal images of individual DNA rings formed at starPEG-(KA7)$_4$ to DNA tile (yellow, Atto633-labeled, $\lambda_{ex} = 640$ nm) ratios from 0.1 to 30. The DNA tile concentration is constant at 50 nM. Scale bar: 2 µm. **d** Histogram of the DNA ring diameters for starPEG-(KA7)$_4$ to DNA nanotube ratios from 0.1 to 30 ($n = (7, 15, 15, 15, 15)$ DNA nanotube rings per condition). Source data are provided as a Source Data file.

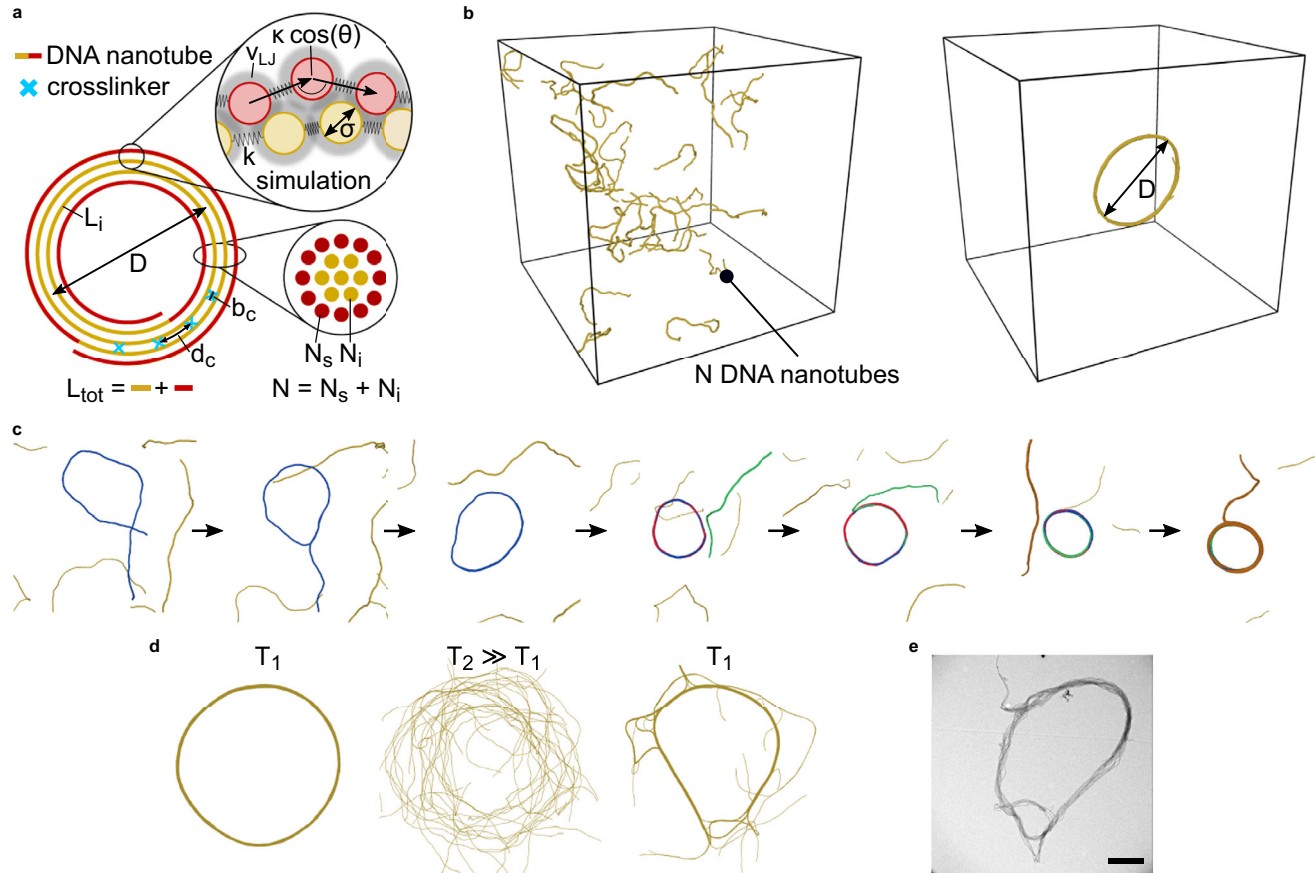

**Fig. 4 | Theoretical description and simulation of DNA nanotube rings.**
**a** Schematic illustration of the theory model representing the associated parameters. The DNA nanotubes are drawn as a continuous line and colored to distinguish surface (red) and interior (yellow) of the ring. The diameter of the bundle cross section is assumed to be negligible compared to the ring diameter $D$. Zoom: A discretized bead-spring representation of the DNA nanotubes is used in the MD simulations with parameters as indicated. **b** Snapshots of an isotropic initialization (left) and a DNA nanotube ring (right) taken from MD simulations. For clarity, the DNA nanotubes widths are increased. The cubic boxes show the simulation volume. **c** Coarse-grained MD simulation of the DNA ring formation from a solution of DNA nanotubes represented as bead-spring polymers. Individual nanotubes involved in ring formation are colored for clarity. A reduced persistence length is employed to facilitate the ring closure. **d** Kinetically trapped structure in incomplete ring formation after simulated annealing (right) (starting at temperature $T_1$ (left) and annealing to a high temperature $T_2 = 8T_1$). **e** Transmission electron microscopy image of a kinetically trapped DNA ring as observed in experiments. Scale bar: 500 nm.

volume of length $L_i$ with a line density of $1/b_c$ of possible binding sites (see Eq. (4) in Supplementary Note 1). In this sense $b_c$ can be viewed as the size of the "footprint" of the crosslinker on the DNA nanotube.

Minimizing the total free energy $F$ with respect to the diameter $D$ at fixed $L_{tot}$, i.e., using $N(D) = L_{tot}/\pi D$, gives the equilibrium diameter of the toroidal DNA nanotube bundle (see Supplementary Note 1). The bending energy will favor large $D$, while adhesion energy and crosslinker entropy favor ring contraction via increasing the overlapping interior length $L_i$ of the bundle. Neglecting prefactors of order unity we find for the equilibrium diameter

$$\left(\frac{D}{L_{tot}}\right)^{(3/2)+\alpha} \sim \frac{L_p}{L_{tot}}\frac{1}{N_c}\left(\frac{gd_c}{k_BT} + \frac{1}{1-b_c/d_c}\right)^{-1}, \quad (3)$$

where $d_c = L_i/N_c \approx L_{tot}/N_c$ is the average distance between crosslinkers (while $b_c$ is their minimal possible distance) and $L_p = \kappa/k_BT$ is the persistence length of individual DNA nanotubes (see Eq. (9) in Supplementary Note 1). This result predicts several experimentally testable scenarios under which ring contraction could occur:

(i) Rings contract for decreasing persistence length or bending rigidity, $D \propto L_p^{1/(3/2+\alpha)}$. If the persistence length $L_p = \kappa/k_BT$ is decreased by increasing the temperature, additional temperature-

dependencies in Eq. (3) become relevant and are discussed under points (iii) and (iv).

(ii) Rings contract if the total DNA nanotube length is decreased, for example, by depolymerization, $D \propto L_{tot}^{(1/2+\alpha)/(3/2+\alpha)}$.

(iii) If crosslinker entropy can be neglected, the ring diameter will contract with increasing $g$ according to $D \propto (\kappa/g)^{1/(3/2+\alpha)}$. This means that if $g$ is an entropic depletion attraction with $g \propto T$ ring contraction occurs for increasing temperature or decreasing persistence length of individual DNA nanotubes, $D \propto T^{-1/(3/2+\alpha)} \propto L_p^{1/(3/2+\alpha)}$ (iii.a). If $g$ is a crosslinker-mediated attraction, it is largely temperature-independent and $D$ is independent of temperature although the persistence length will decrease (iii.b).

(iv) For dominant crosslinker entropy, rings will contract with increasing temperature or decreasing persistence length, $D \propto T^{-1/(3/2+\alpha)} \propto L_p^{1/(3/2+\alpha)}$. The crosslinker entropy is dominant for $gd_c < k_BT/(1-b_c/d_c)$, where $gd_c$ is the average adhesion energy per crosslinker; in particular, it becomes dominant for a dense crosslinker gas with $b_c/d_c \lesssim 1$ if the average adhesion per crosslinker is of the order of several $k_BT$. An additional entropic depletion attraction with $g \propto T$ will further contract rings under these conditions (iv.a).

In summary, the key parameters for ring constriction are $L_p$ and $g$. These can be controlled in simulations as well as experiments with DNA nanotube rings, making the theoretical predictions testable.

## Simulation of the formation and contraction of DNA nanotube bundles and rings

To test the theoretical predictions for ring contraction, we first set up a simulation framework that reproduces the formation of DNA nanotube rings based on coarse-grained MD simulations using LAMMPS[33] with a bead-spring representation of the DNA nanotubes and attraction modeled by a Lennard-Jones potential of strength $\varepsilon$ and particle size parameter $\sigma$ (see Fig. 4a, b and 'Molecular Dynamics Simulation' in the "Methods" section). MD simulations reproduce ring formation from individual DNA nanotubes in solution as shown in Fig. 4c. The lengths of the simulated DNA nanotubes are Poisson distributed, as experiments from previous publications revealed[13,17] and set to a mean value of $580\sigma \approx 6.96\,\mu m$.

Simulations suggest that ring assembly proceeds via nucleation of an initial ring containing few DNA nanotubes and further growth by incorporation of single DNA nanotubes or DNA nanotube bundles.

Sliding of bead-spring polymers in simulations and, thus, equilibration of DNA nanotube rings is impeded by a "lock-in" of beads by attraction to two neighboring beads on a neighboring polymer generating energy barriers for sliding. To facilitate equilibration, we employ an annealing protocol in the MD simulations, where we increase temperature to $T_2 \sim 2-8\,T_1$ for short time intervals for a simulation at temperature $T_1$. Only for very small ring diameters (small $\kappa/\varepsilon$), this annealing procedure is not sufficient to reach equilibrium in available simulation times (three light yellow crosses in Fig. 5c, see Supplementary Note 2 for details). Interestingly, if the annealing temperature $T_2$ is chosen too high, simulated annealing can give rise to kinetically trapped partially unbundled ring structures, which are strikingly similar to experimentally obtained incomplete ring structures observed in TEM (Fig. 4d, e). The simulations can thus reproduce architectural details of the experimentally observed DNA rings, which provides additional validation for our approach (we note that experimental and simulation protocols giving rise to trapped structures can not be compared).

In order to quantify ring diameters of bundles in simulations we measure the 3x3 gyration tensor $S_{mn} = N_{tot}^{-1} \sum_{i=1}^{N_{tot}} r_m^{(i)} r_n^{(i)}$ from all bead positions $\vec{r}^{(i)}$ and obtain a ring diameter by comparing its eigenvalues with corresponding eigenvalues for a homogeneous torus (Supplementary Note 2). We reproduce the theoretically predicted ring contraction in MD simulations. The simulations show contraction both upon decreasing bending stiffness $\kappa$ or persistence length $L_p$ (according to theoretical predictions (i), (iii.a) and (iv), Fig. 5a) and upon increasing $\varepsilon$ or adhesion strength (according to theoretical predictions (iii) and (iv.a), Fig. 5b). Starting from a pre-assembled ring of diameter $D \sim 250\,\sigma$ (corresponding to 3 μm and similar to the rings observed with confocal microscopy) we equilibrate rings over $\sim 10^9$ MD simulation steps, where the ring diameter approaches its equilibrium value. Typically, rings contract during equilibration to diameters in the range of $D \sim 80-250\,\sigma$ corresponding to a reduction of the diameter down to $D \sim 1\,\mu m$ for contracted rings (Supplementary Movies 2 and 3).

We clearly observe smaller equilibrium ring diameters for decreasing $\kappa$ or increasing $\varepsilon$ (Fig. 5a–c) and find a dependence $D \propto (\kappa/\varepsilon)^{2/5}$ characteristic for sliding adhesive DNA nanotubes with $\alpha = 1$ in agreement with the theory result (Eq. (3)) (blue circles and yellow crosses in Fig. 5c). The theory assumes strong interactions deep in the bundled phase. For weak interactions close to the bundling threshold, the bare adhesion energy $g$ should be replaced by the bundling free energy $f = g - Ts$ per length, which is reduced by entropic contributions $s$ from the DNA nanotube shape fluctuations[31]. This gives rise to

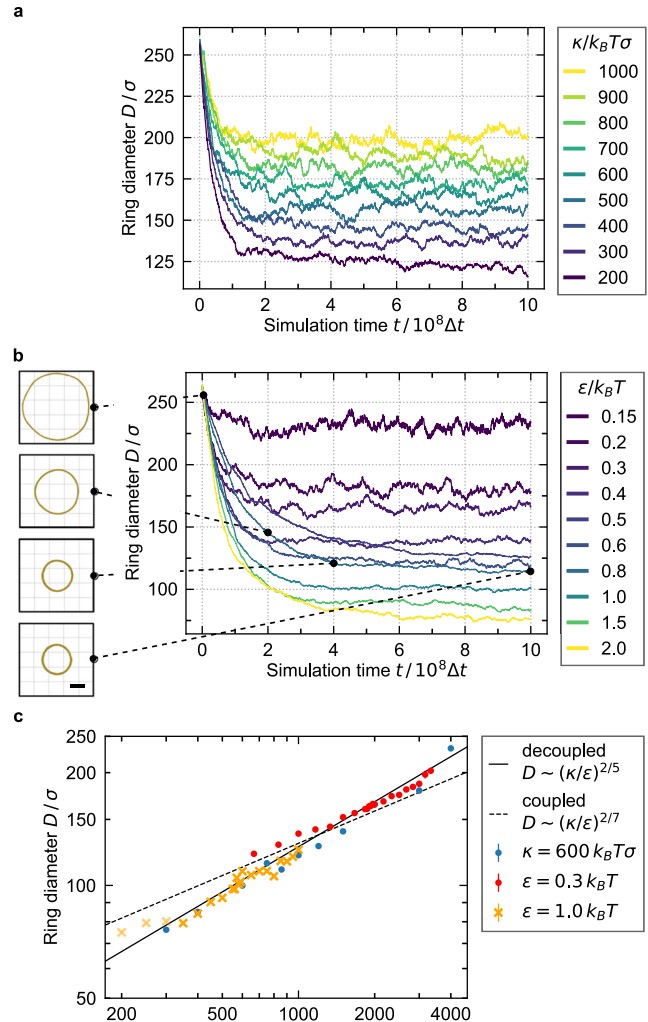

**Fig. 5 | Coarse-grained MD simulations of DNA nanotube rings and their contraction. a, b** Bundle contraction during equilibration (ring diameter as a function of MD simulation time) for different single DNA nanotube rigidities $\kappa$ (**a**, for $\varepsilon/k_BT = 0.3$) and different potential strengths $\varepsilon/k_BT$ (**b**, for $\kappa/k_BT\sigma = 600$). Smaller equilibrium diameters result from decreasing $\kappa$ or increasing $\varepsilon$. Simulation snapshots for $\varepsilon/k_BT = 0.8$. Scale bar: 60 $\sigma$. **c** Double-logarithmic plot of equilibrium ring diameter (Mean ± SD, $n = 545$ measurements, error bars are smaller than symbols) as a function of $\kappa/\varepsilon\sigma$ for increasing bending rigidity $\kappa$ (red circles, for $\varepsilon/k_BT = 0.3$; yellow crosses for $\varepsilon/k_BT = 1.0$) or decreasing potential strength $\varepsilon$ (blue circles, for $\kappa/k_BT\sigma = 600$) in comparison to the theory (3) with $D \propto (\kappa/\varepsilon)^{2/5}$ corresponding to $\alpha = 1$, i.e., sliding decoupled nanotubes (solid line) and $D \propto (\kappa/\varepsilon)^{2/7}$ corresponding to $\alpha = 2$, i.e., shear-resisting coupling between nanotubes (dashed line). Three simulations at smallest $\kappa/\varepsilon$ could not be fully equilibrated (three light yellow crosses). Source data are provided as a Source Data file.

deviations from the proposed scaling (Eq. (3)) for small but realistic values of $\varepsilon$ (red circles in Fig. 5c).

Investigations on a translation of the low viscosities applied in the MD simulation to the more realistic viscosity of water predict an equilibration time of $\sim 20$s (see Supplementary Note 2 for details). These results indicate that ring contraction happens too fast to be observed in our experimental setting because it takes at least half a minute before the microscopy experiment can be started due to mixing of DNA nanotubes with crosslinkers and crowders and filling the solution into an observation chamber. In addition, we can only analyze DNA rings immobilized at the surface which prevents imaging during ring contraction.

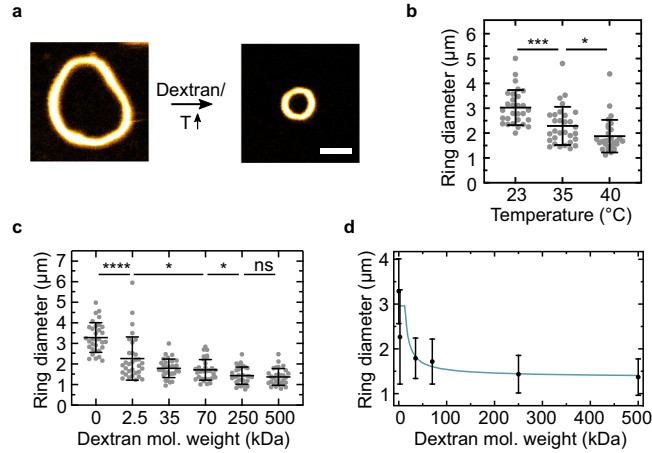

**Fig. 6 | Rings formed from DNA nanotubes contract upon addition of a molecular crowder or heating. a** Confocal images of uncontracted (left) and contracted (right) DNA nanotube rings formed from 50 nM DNA tiles, 500 nM starPEG-(KA7)$_4$ in 1x PBS and 10 mM MgCl$_2$ without and with 25 wt% 500 kDa dextran. Scale bar: 2 μm. **b** DNA nanotube ring diameter after 15 min heating to 35 and 40 °C, respectively (Mean ± SD, $n$ = (31, 30, 30) DNA nantube rings analyzed per condition). Mann-Whitney test with p-values from left to right: ****≤0.0001, *0.0106. **c** DNA nanotube ring diameter for different molecular weights of 25 wt% dextran (2'500, 35'000, 70'000, 250'000, 500'000 g/mol) (Mean ± SD, $n$ = (32, 32, 31, 32, 31, 30) DNA nanotube rings analyzed per condition. Mann–Whitney test with p-values from left to right: ****≤0.0001, *0.0221, *0.0434, ns 0.5787. **d** Experimental DNA nanotube ring diameter reduction by depletion attraction as a function of molecular weight of dextran at 25 wt% (black, identical to data points in **c**, constant total number of dextran monomers). Depletion theory (turquoise, see text and Supplementary Note 3) assumes a penetrable layer of crosslinkers (thickness $p$) around DNA nanotubes (thickness $d_0$). Parameters fitted to experimental ring diameters (black, see **c** (Mean ± SD, $n$ = (32, 32, 31, 32, 31, 30) DNA nanotube rings analyzed per condition)): DNA nanotube diameter $d_0$ = 18.6 ± 2.4 nm, penetration depth $p$ = 11.0 ± 3.3 nm, attraction strength due to crosslinkers $\varepsilon_{cross}/k_BT$ = 0.31 ± 0.14. Source data are provided as a Source Data file.

## Experimental realization of DNA ring contraction

Finally, we set out to realize DNA ring contraction experimentally and use simulations and theory to rationalize the results quantitatively. In experiments, a decrease in persistence length (according to theoretical prediction (i)) can be achieved by increasing the temperature, as temperature is known to decrease the persistence length of DNA[34]. At the same time, a temperature increase leads to nanotube depolymerization, which should cause further contraction (according to theoretical prediction (ii)). We thus form the DNA rings in the presence of starPEG-(KA7)$_4$. Indeed, increasing the temperature from room temperature to 40 °C leads to a reduction of the mean ring diameter from (3.0 ± 0.7) μm to (1.9 ± 0.7) μm (Fig. 6a, b). Note that the ring diameter does not change over time at constant temperature (Supplementary Fig. 9).

To relate the experimentally observed ring contraction to the MD simulations, we have to consider the following dependencies: On the one hand, an increase from room temperature to 40 °C decreases the persistence length of double-stranded DNA by ~10%[34]. On the other hand, we have to consider depolymerization of the free ends of the DNA nanotubes. The critical melting temperature for DNA nanotubes can be calculated as a function of the enthalpy of disassembly, the entropy, the number of sticky end bonds and base pairs per bond and the concentration of free tiles[35,36]. Our DNA nanotube design has a maximum melting temperature of 37.2 °C (at a maximum free tile concentration of 5 nM, compared to 42.0 °C at 50 nM free tiles).

Experiments have shown that in absence of free tiles, the depolymerization rate measures around 0.3 layers per second at 40 °C[35]. Hence, we assume that the free DNA nanotube ends, which we saw with

STED microscopy (Fig. 3b) are depolymerizing at 40 °C, which makes sense if we compare the extent of the ring contraction in the MD simulation to the experiments:

In the MD simulation, the bending stiffness $\kappa$ was decreased from 1000 to $200 k_B T\sigma$ for a fixed attractive strength $\varepsilon$ resulting in smaller ring diameters (see Fig. 5a and red circles in Fig. 5c).

By assigning the temperature-dependent contraction to a purely bending stiffness-dependent contraction we can calculate corresponding effective bending stiffnesses from the simulation data. Measured ring diameters decreasing from 3.0 μm to 1.9 μm (Fig. 6b) for increasing temperatures (23, 35, 40) °C correspond to decreasing effective bending rigidities $\kappa$ = (2010, 890, 500)$k_B T\sigma$ according to the simulation data for a fixed attractive strength of 0.3$k_B T$ (red circles in Fig. 5c). This predicted four-fold decrease in effective bending stiffness is not sufficient to explain the experimentally observed ring contraction. It confirms that both, a decrease of the persistence length and depolymerization reduce the effective ring parameter in experiments, validating theoretical predictions (i) and (ii). Indeed, depolymerization is also relevant for the contraction of actin rings[4,37].

To test theoretical predictions (iii) and (iv), we need to increase $g$ with an additional depletion force. We can achieve this experimentally by adding dextran as a molecular crowding agent, which allows us to control the depletion attraction with the concentration and the molecular weight of the crowding agent. We thus add 25 wt% of dextran to the DNA ring-containing solution. We find that the additional molecular crowding induces DNA ring shrinkage to less than 45% of the initial diameter from a mean diameter of (3.3 ± 0.7) μm to (1.4 ± 0.4) μm at constant temperature (Fig. 6c), confirming theoretical prediction (iii). Importantly, the rings' shape, quantified by their circularity, remains unaltered and close to 1 (circularity of 1 corresponds to a perfect circle, Supplementary Fig. 10). We observe similar results with methylcellulose, confirming that the contraction is induced by crowding and not due to the chemical nature of the agent (Supplementary Fig. 11). The high uncertainty of ring diameters might mainly result from highly heterogeneous bundles that consist of DNA nanotubes of different lengths (Poisson-distributed). Overall, we observe that a higher molecular weight of dextran gives rise to smaller ring sizes, i.e., an effectively increased attractive interaction (theoretical prediction (iv.a)).

This ring contraction can be understood quantitatively by calculating the additional depletion interaction which arises from exclusion of dextran from an overlap volume between DNA nanotubes. Details of the calculation can be found in Supplementary Note 4. The molecular crowding effect of a macromolecule is, on the one hand, dependent on its radius of gyration (and thus on its molecular weight), which sets the thickness $\delta$ of the depletion layer (the depletion length) and, on the other hand, on its concentration. It is important to state that the number concentration of dextran molecules in the experimental sample decreases with higher molecular weights if the mass concentration is kept constant (and thus the number of monomers per sample). For hard rods of diameter $d$, the combined effects of an increasing radius of gyration $R_g \propto M^{1/2}$ of the molecular crowder (increasing the depletion layer thickness), and a decreasing number concentration $c/M$ (at fixed mass concentration $c$) give rise to a depletion attraction that decreases with molecular weight $\propto M^{-1/2}$ for small radii of gyration $R_g \ll d$ before it saturates for large radii of gyration $R_g \gg d$ corresponding to large molecular weights $M$. The model of a hard rod is, however, not completely adequate in the presence of additional crosslinkers, which "decorate" the DNA nanotubes and act as a penetrable layer of thickness $p$ around the DNA nanotubes of bare diameter $d_0$. The glycocalix around red blood cells constitutes a similar penetrable layer that has been shown to give rise to increased aggregation of red blood cells by dextran of higher molecular weight[38]. Following Neu et al.[38], we treat the crosslinker-decorated DNA nanotubes as penetrable rods of total diameter $d = d_0 + 2p$ with a reduced

depletion length $\delta - p$. The depletion layer thickness $\delta$ is reduced by a penetration depth $p$, which is the thickness of the penetrable layer: a part of the crowding agent of size $p$ can be "buried" within the penetrable layer, which reduces the size of the depletion layer accordingly. As a result, smaller crowding agents with less molecular weight become less effective. For $p$ as small as a few nanometers, this gives rise to a depletion attraction $\varepsilon_{dep}(M)$ that increases with molecular weight $M$ for $R_g \ll d$ before it saturates for high molecular weight ($R_g \gg d$). Using the ring diameter result $D \approx 100 d_0 (\varepsilon/k_B T)^{-2/5}$ from our MD simulations with an attractive strength $\varepsilon = \varepsilon_{cross} + \varepsilon_{dep}(M)$ that is the sum of a crosslinker contribution and the depletion contribution, we can fit the experimental data for the ring diameter as a function of dextran molecular weight quantitatively (see Fig. 6d, blue line) with fit parameters $d_0 = 18.6 \pm 2.6$ nm and $p = 11.0 \pm 3.3$ nm for the bare DNA nanotube diameter and the penetration depth, respectively. A DNA nanotube diameter of $11.8 \pm 2.1$ nm has been measured by TEM (unbundled DNA nanotubes in the absence of starPEG-(KA7)$_4$, Supplementary Fig. 4). The size of the starPEG-(KA7)$_4$ crosslinkers can be estimated to be below 22 nm (see above and earlier publications[26]): These values are compatible with the fit results in view of additional complications, such as tilted orientations of attached crosslinkers and the possibility that the KA7 peptides align due to their electrostatic interactions in parallel with the DNA nanotube, which reduce the value of $p$. The fit from the MD simulation reproduces the experimental data qualitatively as well as quantitatively.

With temperature increase and the addition of molecular crowders, we were able to obtain contraction of the DNA nanotube rings to less than half of their initial diameter. With this, we validate the four theoretical predictions, which we used as guidance to derive the parameters relevant for ring contraction. Our DNA rings thus contract significantly more than rings formed from actin filaments when they are contracted by passive crosslinkers[4], likely due to the lower persistence length of the DNA nanotubes compared to actin.

In conclusion, we engineered synthetic micron-scale DNA rings, self-assembled from a bundle of tens of DNA nanotubes and crosslinked via electrostatic interactions with custom-designed starPEG-(KA7)$_4$ peptides. Based on theoretical considerations, we derived conditions for DNA ring contraction, which we validate with experiments and coarse-grained MD simulations. Since our micron-sized DNA rings consist of well-established DNA nanotubes, further investigations, e.g. involving mutations like stiffness control[39,40] are facilitated compared to protein-based materials like actin filaments. In the future, these micron-sized DNA rings could be used as tracks for molecular assembly and transport or embedded into adaptive materials. In addition, synthetic DNA rings are already equipped with features responding to temperature change similar to their biological counterparts (polymerization of actin filaments[41] and DNA nanotubes[35,36]). By contrast, DNA nanotubes can easily be equipped with other types of molecular functionalization, reprogrammed, and repurposed for diverse systems.

An entirely synthetic division machinery for liposomes, based on DNA nanotechnology and peptide design, is a highly attractive albeit far-reaching goal. It has to be acknowledged that besides complete contraction several challenges have to be overcome to induce vesicle division due to membrane fission. The rings have to be positioned in the equatorial plane of the liposome, which will likely be achievable by self-assembly if the persistence length of the DNA nanotubes is sufficiently high[13]. Secondly, the rings have to be linked to the membrane which could be achievable e.g. with cholesterol-tags[13] or transmembrane entities[15]. The temperature-induced contraction would in principle be compatible with liposome-encapsulation and the contraction force could potentially be sufficiently high to induce vesicle

deformation[26]. Our approach could be complemented by engineered molecular motors that walk on DNA nanotubes[16]. For ring disassembly, which will be necessary to complete the division of the compartment, it is plausible to use mechanisms that have already been described for DNA nanotubes[12,13,17]. Each of these steps, however, warrants detailed investigation and presents a fruitful challenge for future research.

The symbiosis of DNA nanotechnology and peptide engineering may lead to advanced and highly functional molecular hardware for bottom-up synthetic biology and hybrid materials with a wide range of applicability.

## Methods

### DNA nanotube design and assembly

DNA nanotube sequences were adapted from the original single-tile design by Rothemund et al.[11]. Each tile is composed of five DNA oligomers, the DNA sequences are listed in Supplementary Table 1. The five DNA oligomers were mixed to a final concentration of 5 μM in 1× phosphate-buffered saline (PBS, pH 7.4) and 10 mM MgCl$_2$. The tiles were annealed using a thermocycler (Bio-Rad) by heating the solution to 90 °C and cooling it to 25 °C in steps of 0.5 °C for 4.5 h. The assembled DNA nanotubes were stored at 4 °C and used within two weeks. The DNA oligomers were purchased from Integrated DNA Technologies or Biomers (purification: standard desalting for unmodified oligomers, HPLC for oligomers with biotin and fluorophore modifications) and stored at 100 μM in 1 × Tris-EDTA (pH 8) and stored at −20 °C.

### Peptide synthesis

The synthetic peptides, KA7, starPEG-(DA7)$_4$ and starPEG-(KA7)$_4$, were synthesized as described in detail in Drechsler et al.[26]. Briefly, all peptides were prepared by a standard Fmoc solid phase synthesis approach on an automated solid-phase peptide synthesizer (ResPep SL, Intavis) using 2-(1H-benzotriazole-1-yl)-1,1,3,3-tetramethyluronium hexafluorophosphate (HBTU) activation. Each amino acid was coupled twice with fivefold excess, while all non-reacted amino groups were capped with acetic anhydride. Likewise, 5(6)-TAMRA was coupled to the N-terminus of the peptides still bound the resin. Peptides were removed from the resin with trifluoroacetic acid/triisopropyl silane/water/dithiothreitol (DTT) (90[vol/vol]:5[vol/vol]:2.5[vol/vol]:2.5[m/vol]) for 1.5 h and precipitated with ice-cold diethyl ether. Peptides were further purified by reverse-phase high-pressure liquid chromatography (HPLC, Waters, Milford, MA) on a preparative C18 column (AXIA 100A, bead size 10 μm, 250 × 30 mm, Phenomenex, Torrance, CA) and analyzed by electrospray ionization mass spectrometry (ACQUITY TQ Detector; Waters) for purity. Cysteine-terminated KA7 and DA7 peptides were coupled to maleimide-terminated starPEG (10 kDa) by Michael addition reactions. For this, peptides and starPEG were mixed at a 1:5 (star-PEG:peptide) molar ratio in 1× PBS (pH 7.4), sealed and stirred over night at 750 rpm and room temperature. The resulting star-PEG peptides were dialysed for 2 days against water using tubing with an 8-kDa cut-off. Peptide products were lyophilized and stored at −20 °C. For experiments, the respective peptides were rehydrated in 1× PBS (pH 7.4) and stored as single-use aliquots at −20 °C. Peptide aliquots were stored at 1 mM and −20 °C. For experiments they were diluted in 1× PBS and used within one day. The starPEG construct synthesis is described in greater detail in Wieduwild et al.[42]. The used peptide has been characterized by mass spectroscopy by Drechsler et al.[26] and the starPEG-peptide hybrid by NMR by Thomas et al.[43].

### Confocal fluorescence microscopy

A confocal laser scanning microscope LSM 900 (Carl Zeiss AG) was used for confocal microscopy. The pinhole aperture was set to one Airy Unit and the experiments were performed at room temperature (unless stated otherwise). The images were acquired using a 20× (Plan-

Apochromat 20×/0.8 Air M27, Carl Zeiss AG) or 63× objective (Plan-Apochromat 63×/1.4 Oil DIC M27). Images were analyzed and processed with ImageJ (NIH, brightness and contrast adjusted, ImageJ 2.3.0/1.5q; Java 1.8.0_322 64-bit,[44]).

## STED imaging

DNA rings were imaged on an Abberior expert line (Abberior Instruments GmbH, Germany) with a pulsed STED line at 775 nm using an excitation laser at 640 nm and spectral detection. Detection windows was set to 650–725 nm to detect Atto633-labeled DNA nanotubes. Images were acquired with a 100×/1.4 NA magnification oil immersion lens (Olympus). The pixel size was set to 30 nm and the pinhole was set to 1AU. Images were analyzed and processed with ImageJ (NIH, brightness and contrast adjusted).

## Charge, crowding, and multivalency control assay

In order to compare the effect of different synthetic peptides on the DNA nanotubes, 30 nM DNA nanotubes were incubated with 500 nM positively charged multivalent starPEG-(KA7)$_4$-TAMRA, 500 nM of the negatively charged multivalent starPEG-(DA7)$_4$-TAMRA and 2 µM of the the positively charged monovalent (KA7)$_4$-TAMRA or observed in absence of synthetic peptides. Buffer solutions contained 1× PBS and 10 mM MgCl$_2$. Samples were imaged in a custom-made observation chamber after one hour of incubation at room temperature (if not stated otherwise). The laser settings were kept the same at all time points. We analyzed the mean intensity per pixel in the same manner as we did for the colocalization assay. For each condition, nine overview images were analyzed, single data points and Mean ± SD were plotted with GraphPad Prism (Version 9.2.0 (332)).

## Colocalization assay

For the preparation of DNA nanotubes A (green), the composition of DNA oligomer mix was altered to 90% 6-FAM-labeled SE3 strand and 10% biotinylated SE3 strand (Supplementary Table 1). DNA nanotubes A were annealed as described above. 10 mg neutravidin was dissolved in 1 mL ultrapure water and then further diluted in 1× PBS to a final concentration of 0.2 wt%. Biotinylated-BSA (Thermo Fisher) was diluted in ultrapure water to a concentration of 0.2 wt% and diluted in 1× PBS to a final concentration of 0.1 wt%. The bottom glass slide of a custom-built observation chamber was coated with 0.1 wt% biotinylated BSA for 15 min. First, 50 µL of 0.2 wt% neutravidin were added on top of the slide and incubated for 2 min. After that, it was washed with 1× PBS. Subsequently, 30 nM 6-FAM-labeled biotinylated DNA nanotubes A in 1× PBS and 10 mM MgCl$_2$ were added. To seal the observation chamber, the upper slide was placed on top, spaced by double sided tape and sealed with removable two component glue. After 20 min incubation, the chamber was reopened and 50 µL of 5 µM starPEG-(KA7)$_4$ diluted in 1× PBS was flushed. After an incubation of 2 min, 50 µL of 50 nM DNA nanotube B (green, Atto633-labeled) and diluted in 1× PBS was flushed and washed again with 1× PBS. The observation chamber was sealed with two component glue for imaging. Images were acquired with confocal microscopy as described above.

For the starPEG-(KA7)$_4$ colocalization assay increasing amounts of starPEG-(KA7)$_4$-TAMRA were added to the DNA nanotubes. The mix contained starPEG-(KA7)$_4$-TAMRA at varying concentrations (0, 50, 100, 200, 500 nM), 30 nM DNA nanotubes, 1× PBS and 10 mM MgCl$_2$. The respective mix was left to incubate at room temperature for one hour. We then imaged the samples at the confocal laser scanning microscope with the ×63 oil objective, keeping the same laser settings for all conditions.

We then analyzed the mean pixel colocalization intensity $c_p$ of TAMRA-labeled peptides at the position of the DNA nanotubes as follows. The analysis was performed with ImageJ (ImageJ 2.3.0/

1.5q; Java 1.8.0_322 64-bit,[44]) and the plugin Skeleton. The pixel intensities for both images of either the DNA nanotube channel or the TAMRA channel range from $i = 0$ to 255 (Supplementary Fig. 2a,d). In order to extract the information about the DNA nanotube bundle positions, we first adjusted the threshold using the Method 'Otsu', thereby creating a binary image (Supplementary Fig. 2b). In the next step, we used the plugin skeleton to reduce the DNA nanotube bundles to one single row of pixels in the middle of the DNA nanotube bundle, so we could generate a mask to always analyze the same area per DNA nanotube length unit (Supplementary Fig. 2c). Then we linked the binary skeletonized information on location with the picture in the TAMRA-labeled channel by using the ImageCalculator function 'AND'. Thereby we obtain background pixels counting zero intensity ($i = 0$) and skeletonized DNA nanotube (-bundle) pixels with non-zero intensity ($i = 1–255$; Supplementary Fig. 2e). The latter are included in the calculation of the average pixel intensity, called colocalization intensity. The resulting histogram carries the pixel number $p_i$ for each intensity value $i$. The pre-processing of the images directly leads to the fact that pixels with the intensity value zero are not included in the calculation of the average intensity at the DNA nanotube(-bundle) centers. The following equation outlines the relation.

$$\overline{c_p} = \frac{\sum_{i=1}^{255} i \cdot p_i}{\sum_{i=1}^{255} p_i} \qquad (4)$$

We calculated the mean of the pixel colocalization intensity $c_p$ and its standard deviation using ten overview images per sample. For 0 nM starPEG-(KA7)$_4$ we analyzed five overview images. The plot was generated via GraphPad Prism (Version 9.2.0 (332)). The same overview images were used to manually count the number of DNA rings.

## Transmission electron microscopy

For negative staining, 10 µL of DNA nanotube-containing solution (1 × PBS, 10 mM MgCl$_2$, 30 nM DNA nanotubes, and starPEG-(KA7)$_4$ concentrations as described) 0.1% paraformaldehyde was applied onto a glow-discharged 100 mesh copper grid with carbon-coated Formvar (Plano GmbH) and incubated for 30 min under a box. The solvent was removed by gentle blotting from one side with filter paper. The grid was rinsed with 3 drops of water, blotted again, and treated with 10 µL of 0.5% (w/v) uranyl acetate solution for 20 s. After removing the staining solution thoroughly by blotting with filter paper, the grid was air dried and imaged on a FEI Tecnai G2 T20 twin transmission electron microscope (FEI NanoPort) operated at 200 kV. Electron micrographs were acquired with an FEI Eagle 4k HS, 200 kV CCD camera at 14,500x nominal magnification.

## Bundle thickness analysis

In all, a mixture of 30 nM DNA nanotubes with 0, 25, 50, 200, and 500 nM starPEG-(KA7)$_4$ was prepared in 1× PBS containing 10 mM MgCl$_2$. Mixed samples were incubated at room temperature for 30 min and prepared for transmission electron microscopy. Images were acquired as described above at 14500x nominal magnification. Images were imported in ImageJ (ImageJ 2.3.0/1.5q; Java 1.8.0_322 64-bit,[44]) and the bundle thicknesses were evaluated by taking line profiles at manually selected positions for the DNA nanotube bundles. Per condition 100 positions were randomly chosen and measured. The data was plotted with GraphPad Prism (Version 9.2.0 (332)) as Mean ± SD.

## StarPEG-(KA7)$_4$-assisted DNA ring self-assembly, ring size control assays, and ring size measurements

The starPEG-(KA7)$_4$ stock solution (1 mM) was prediluted to 5 µM in 1× PBS. The according amount of starPEG-(KA7)$_4$ was added to the DNA nanotube mixture containing a final concentration of 50 nM DNA

nanotubes in 10 mM MgCl$_2$ and 1× PBS (if not stated otherwise). The starPEG-(KA7)$_4$:DNA tile ratio was set to 10 (if not stated otherwise). For colocalization experiments (Figs. 1d, 2b–d and Supplementary Figs. 2, 3, and 5) and self-assembled ring observations (Fig. 3, Supplementary Figs. 6–11) 20 µL solution were added to a custom-built observation chamber. Self-assembled rings immediately stick to the uncoated glass slide due to the negative charge of DNA and are thereby immobilized for imaging.

For the ring size control with crowding molecules (Fig. 6c), 25 wt% dextran (2500, 35,000, 70,000, 250,000, 500,000 g/mol) was added to the DNA nanotube-starPEG-(KA7)$_4$ mixture immediately after preparing the latter. Per condition, 100 µL of sample solution were prepared and entirely pipetted into a well slide (ibidi µ-Slide 18 Well, glass bottom) and imaged without delay at the confocal laser scanning microscope (Carl Zeiss AG) using the ×63 oil objective. The laser settings were identical for all images.

For the ring size control with temperature increase (Fig. 6b), 500 µL of sample were prepared and at each time step 100 µL were pipetted into a well slide (ibidi µ-Slide 18 Well, glass bottom) inserted into a heating chamber (ibidi Temperature Controller, blue line). Lid and plate were set to the temperature of interest and the remaining sample was incubated at the same temperature for 15 min prior to pipetting.

For all ring size control experiments, 30 images of single rings were taken (5x zoom) per condition. Images were analyzed with ImageJ (ImageJ 2.3.0/1.5q; Java 1.8.0_322 64-bit,[44]) by tracing single rings with the freehand ROI selection. Circularity values were measured using the shape description feature of ImageJ. The perimeter $p$ was measured, from which the diameter $d$ was deduced by $d = p/\pi$. Thereby ring shapes that deviate from the circular were approximated to a circle and the ring size is expressed as the according diameter of an exact circle. Data was plotted as Mean ± SD with GraphPad Prism Version 9.2.0 (332).

## Molecular dynamics simulation

Coarse-grained MD simulations are performed using LAMMPS[33] with a Langevin dynamics thermostat in order to obtain a realistic dynamics without explicit solvent. DNA nanotubes are represented as bead-spring polymers consisting of $N_b$ beads with diameter $\sigma$ connected by stiff harmonic bonds with rest length $\sigma$ and with a bending rigidity $\kappa$ resulting in a mean contour length $\langle L \rangle \approx N_b \sigma$ and a persistence length $L_p = \kappa/k_B T$. The attraction between DNA nanotubes is modeled by a Lennard-Jones potential of strength $\varepsilon$ and particle size parameter $\sigma$, i.e., we do not simulate explicit crosslinkers but a ring contraction due to a generic DNA nanotube attraction corresponding to $g \approx \varepsilon/\sigma$.

Typical experimental parameters for DNA nanotubes correspond to a length-to-diameter ratio of $N_b = \langle L \rangle/\sigma \approx 580$ with a DNA nanotube diameter $\sigma \approx 12$nm and to a contour to persistence length ratio $\langle L \rangle/L_p \approx 1$ or $\kappa \approx N_b k_B T \sigma$. The bending rigidity is reduced to $\langle L \rangle/L_p \approx 3$ in simulations of the ring formation to observe this process on computationally accessible time scales (Supplementary Note 2). We employ a cubic simulation box of side length $800\sigma$ with periodic boundaries and typically simulate 30 DNA nanotubes (consistent with the range of DNA nanotubes within a bundle as extracted from electron microscopy images) corresponding to a total number $N_{tot} \approx 17400$ beads.

For ring contraction to their final equilibrium diameter, we perform $\sim 10^9$ MD steps, but employing the Langevin thermostat with a viscosity much lower than realistic viscosities of water, $\eta \approx 10^{-5}\eta_{Water}$, for faster equilibration. This corresponds to simulation times $\sim 20$ ms for such low viscosities. Because equilibration time depends linearly on viscosity in the overdamped regime and is essentially independent of viscosity in the underdamped regime, and the crossover between both regimes happens for

$\eta \approx 10^{-3}\eta_{Water}$, this corresponds to equilibration time scales $\sim 20$ s if the viscosity of water could be used (see Supplementary Note 2 for details). This suggests that ring contraction happens on the time scale of several seconds, which is difficult to follow directly in experiments (mixing and pipetting into an observation chamber until the start of imaging takes at least half a minute).

The ring diameters shown in Fig. 5c are computed as the mean diameter for simulation times $t/\Delta t > 8 \times 10^8$ to exclude the equilibration process. This measurement time frame is shifted to $t/\Delta t > 18 \times 10^8$ for simulations with $\varepsilon/k_B T = 1.0$ and ratios $\kappa/\varepsilon\sigma \leq 400$ taking into account the prolonged equilibration time (see Supplementary Note 2). Measurements are not performed at $T_2$, i.e., during annealing, but only after switching back to $T_1$ and waiting for at least $5 \times 10^4$ time steps. Supplementary Note 2 contains further information on how the ring diameters are calculated from the gyration tensor and on the equilibration of ring diameters.

## Statistical analysis

No statistical method was used to predetermine sample size. All the experimental data were reported as Mean ± SD from $n$ experiments, DNA nanotubes or rings. The respective value for $n$ is stated in the corresponding figure captions. To analyze the significance of the data, a Student's t-test (unpaired, nonparametric, Mann–Whitney test) with two-tailed $p$-values was performed using GraphPad Prism (Version 9.2.0 (332)) and $p$-values correspond to ****$p \leq 0.0001$, ***$p \leq 0.001$, **$p \leq 0.01$, *$p \leq 0.05$ and ns: $p \geq 0.05$. Experiments that revealed micrographs in Figs. 1d, 2b–c, 3a–c, 4e, and 6a have been repeated at least twice and Fig. 2b–c have been recorded once.

## Reporting summary

Further information on research design is available in the Nature Portfolio Reporting Summary linked to this article.

# Data availability

The datasets generated during and analyzed during the current study are available in the repository called "Triggered contraction of self-assembled, micron-scale DNA nanotube rings [Research Data]" on heiDATA/Göpfrich Group - Biophysical Engineering of Life with the identifier https://doi.org/10.11588/data/ADYUNN[45]. Source data are provided with this paper.

# Code availability

The code that supports the findings of this study is available in https://doi.org/10.5281/zenodo.10728688.

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

## Acknowledgements

We thank Elisa D'Este and the Optical Microscopy Facility at the Max Planck Institute for Medical Research for their assistance with STED imaging. K.G. acknowledges funding from the Deutsche Forschungsgemeinschaft (DFG, German Research Foundation) under Germany's Excellence Strategy via the Excellence Cluster 3D Matter Made to Order (EXC-2082/1 – 390761711) and the ERC starting grant "ENSYNC" (No. 101076997). M.I. and K.G. thank the Hector Fellow Academy. K.J. thanks the Carl Zeiss Foundation, the Joachim Herz Foundation and the Alexander von Humboldt Foundation for financial support. All authors acknowledge the Max Planck Society for its general support.

## Author contributions

M.I., K.J. and L.P.W. contributed equally. K.G. and S.D. conceived and designed the experiments. J.K. and L.P.W. conceived the

theoretical analysis. M.I. developed and M.S. conducted bundling experiments. M.I. performed the colocalization assay. M.S. analyzed the colocalization assay data. M.S. performed ring yield experiments. M.I. and M.S. performed ring-size experiments. M.I. analyzed the bundle thickness and ring size and shape data. K.J. supervised M.I. and M.S. and M.I. supervised M.S. in the lab. U.M. performed electron microscopy imaging. Y.Z., H.D. and S.D. contributed and characterized the starPEG and peptide constructs. L.P.W. performed and analyzed simulations. K.J., M.I., L.P.W., J.K. and K.G. wrote the manuscript with the help of all authors.

## Funding

## Competing interests

The authors declare no competing interests.
