## [Peer Review File · Nature Communications]

REVIEWER COMMENTS

Reviewer #1 (Remarks to the Author):

In their manuscript, Illig & al. present a novel DNA nanotechnology system consisting of DNA nanotubes that self-assemble from DNA tiles into micron-sized rings. The motivation and goal of this work are to achieve a division mechanism for systems such as synthetic cells, which is an in-vogue synthetic biology field of research. The results shown in this manuscript demonstrate that DNA nanotechnology could be a new route towards this goal, albeit it would require additional extensive work.

The DNA nanotubes, in the presence of a multivalent positively charged block co-polymer (starPEG) made of a four-armed PEG core and positively charged short Lysine-Alanine peptide repeats, further self-assemble into bundles that can loop onto themselves and form rings. They first show that the formation of those rings is due to the multivalent electrostatic interactions between the DNA tiles self-assembled into nanotubes, and the positive charges on the four-armed starPEG cross-linkers. On the one hand, they show that the diameter of those rings is independent of the concentrations of DNA tiles - provided that a ten-fold molarity of starPEG is present - and of the magnesium ions. On the other hand, they show that this system is stable with time. They provide a theoretical frame that describes the formation and the time evolution of the DNA rings by considering them as a single DNA nanotube coiled multiple times around itself, that traps a gas of crosslinkers in its interior. The model takes into account the bending rigidity of the coiled DNA nanotube, an adhesion force between the different coils of the DNA nanotube in the interior of the DNA ring, and the free energy of the gas of crosslinkers. Using this model, theoretical predictions are made with respect to the dependence of the diameter of the DNA ring on the bending stiffness of the DNA coil and the adhesion force between the coils. The predictions about the bending stiffness of the DNA coil are tested experimentally by changing the temperature of the system, while the ones related to the adhesion force are tested experimentally by adding a depletion-attraction force mediated by a molecular crowder consisting of Dextran of increasingly large molecular weights. They successfully report the decrease in the diameter of those DNA rings as they increase the temperature of the system or add molecular crowders, consistent with the predictions of their model.

Overall, the work shows an interesting first step towards achieving a division mechanism based on DNA nanotechnology. The manuscript is clear and concise for the most part. The experiments are well described and illustrated by clear figures and provide a convincing characterization of the DNA rings. The conclusions and perspectives about the extent of the work are honest.

The description of the model is not as clear. A figure would be very helpful to give a visual representation of the different parameters that describe the DNA ring and the interactions that take place.

Although a mapping of the lengths between simulations and experiments is provided, none is given for the time. It would be interesting to give the simulation time Δt in actual time and compare the dynamics obtained from simulations to those observed in experiments. The duration of annealing used in MD

simulations could, for example, also be compared to the one used in the synthesis of the nanotubes. A brief comment about the choice of T_2/T_1 with respect to ϵ would be useful.

In Figure 4, for the larger value of potential strength (yellow crosses), why is there a step of the value of the diameter at lower values of the stiffness? That deviation from the provided scaling law should be commented on.

Some typos:

Line 131, blank space missing before "StraPEG"

Figure 4e) caption, blank space is missing in "decreasingpotential"

Line 309, $R_g \ll d$ to replace with $R_g \gg d$

Line 450, a period is missing before "The pre-processing"

Reviewer #2 (Remarks to the Author):

This is a nice paper reporting novel and potentially useful information on the synthetic contractile rings. In this paper, the authors summarize important aspects related to the formation of micrometer-scale rings using DNA nanotubes and their contraction, which are crucial for creating artificial cells. The main results they presented are as follows:

- Authors demonstrated the formation of micrometer-scale DNA nanotube rings that are crosslinked by starPEG-peptides.
- The bundling of DNA nanotubes by starPEG-peptides is due to crosslinking through their charge, not depletion force.
- Based on the theory that crosslinkers maximize the overlap region between filaments due to entropic force, authors described the parameters and characteristics of the DNA rings.
- Authors reproduced the contraction process of the ring based on the theory using CG-MD simulation.
- Furthermore, authors confirmed the contraction of the ring by temperature changes and addition of molecular crowders, based on the prediction from the theory.

Overall, it is novel and surprising that micrometer-scale rings can be formed using DNA nanotubes and that they are contracted to less than half of their original size. The process of verifying the characteristics predicted by theory and MD through experiment is impressive. However, I believe some concerns should be addressed before publication.

The only major criticism I have is that the advantages of using DNA nanostructures are not maximized, and there seems to be a lack of description or experimentation. Why use DNA instead of naturally-occurring actin filaments? The advantages of using DNA nanotubes include controllable stiffness, easy functionalization, sequence addressability, and application of powerful genetic tools. If such experiments are not feasible, it would be helpful to elaborate further on the advantages of using DNA nanotubes in artificial cells for bottom-up synthetic biology. I understand the power of DNA nanotechnology and that references such as 12–16 and 20–24 are already listed in the introduction section. However, most readers may not have a clear picture, so highlighting the specific advantages of using DNA nanotubes over cytoskeletal filaments for future artificial cells would help readers understand the importance of this paper.

Minor points:

- Line 16 in Abstract: I found it somewhat difficult to understand what the authors refer to by “these two conditions.” It may be clearer if the clause in the previous sentence is changed to, for example, “--upon (i) increasing attraction or (ii) decreasing bending rigidity of the DNA nanotubes, ...”
- The length distribution of DNA nanotubes seems to be missing. Does the initial length of DNA nanotubes affect the formation and contraction of the ring? How the experimentally obtained length of nanotubes is implemented in the simulation?
- Fig. 3b: The description “Representative confocal (left) and STED (right) images” should be replaced with “Representative confocal (top) and STED (bottom) images” ?
- Fig. 5d: Is this the result of MD simulation? not experiment?
- Line 205: There is no “iv.b”, so “iv.a” is not required?
- Line 165: The technical term ‘gas’ may be a common term among theoretical people, but I feel that it would be helpful to provide an explanation for ease of reading. I should have read some previous papers to find out what exactly ‘gas’ means.
- Line 347: I was wondering if the reference number is correct: “Our approach could be complemented by engineered molecular motors that walk on DNA nanotubes[37].”

Reviewer #3 (Remarks to the Author):

Illig et al. present an experimental study, supported by basic theory and coarse-grained simulation, of contraction of synthetic DNA rings. The system is intended to mimic contraction of cytoskeletal rings,

which play an important role in processes such as cell division. The authors demonstrate that they can achieve impressive degrees of contraction by either adding molecular crowders or temperature increase. Qualitatively, this behaviour is consistent with the predictions of the simulation and the basic theory. Overall, the system is interesting and the evidence persuasive. The manuscript is also well written. I am therefore inclined to recommend publication, subject to addressing the technical criticisms below.

Please note that I am not an expert in the experimental methods here and I am not in a good position to comment on their reliability. The approaches seem reasonable to me, and plenty of detail is given.

Comments:

1. It wasn't that clear to me what the authors want the overall interplay between the theory, simulation and experiment to be. Is the idea that the simulations test whether the basic theory is reasonable; having done so, this basic theory is then used to select certain triggers for forcing the system to contract? The authors don't quite build enough of a narrative to hold these three parts together. Perhaps simply stating more explicitly how each part of the work relates to the overall goal would help. For example, how does fitting the scaling of the simulation data to the theory in 4e help with the overall goal?
2. A lot of the figures are very small. Is this really necessary?
3. In some cases it isn't immediately obvious what the reader should take from the figures. Eg. Fig 2 - what do I learn from 2b? What is it about the figure that tells me this?
4. I'm unclear on the purpose of the exponential fit in 2d and 2f. Is there a theory that says an exponential is expected? Otherwise I would not recommend fitting (although a spline to guide the eye may be reasonable).
5. In the discussion of the ring geometry, I was initially very confused because I was expecting some discussion of what drives the ring to form in the first place. In fact, I think the authors start from the assumption that rings have formed and then discuss how the geometry changes in response to the environment. It would be helpful to make this clear up front.
6. The gas gives an "additional" effective adhesion. Additional to what? Are the authors saying that the crosslinker have two contributions, one of which is this entropic term?

7. I would recommend having a figure in the main text to illustrate the theoretical concepts discussed on p9-11, and the features of the CG model, schematically. It would clarify a lot of what is currently discussed only in words, with terms like "overlap" that are not formally defined.

8. What do the colours mean in Fig. 3e?

9. I think it's worth saying that the model assumes uniform bending in a filament of uniform susceptibility.

10. I initially thought D was the diameter of the cross-section of the bundle; it may help to be more explicit.

11. What does b_c describe in the actual physical system? Is it the size of the cross-linker's footprint?

12. The derivations performed using the model (reported in SI) should be explicitly referenced in the text. For example, where is Eq. (3) derived?

13. I found the $/2+\alpha$ notation slightly ambiguous. I don't think 2 and α are supposed to be in the same denominator, but it might be misinterpreted that way.

14. The actual definition of the CG model is not very explicit in/explicitly referenced from the main text. The schematic diagram would help here, as it would help to explain the attraction to "two neighbouring beads", which I don't think is hard-coded but arises from the way the chains fit together.

15. Temperature ranges of $2-8 T_1$ are not "comparable to typical experimental" protocols, wherein the temperature typically changes by 25% or less.

16. Fig 4a isn't well described by the caption. What are all the subfigs showing? I think only the third subfig is "trapped".

17. Given the importance of temperature scaling, it would be helpful to state which parameters are assumed to be temperature-independent (eg. κ) when they are introduced.

18. Top of page 16: paragraph starts with "to compare", but I don't think a comparison follows.
19. On p16, the authors talk about a melting temperature "below 5nM" of free tile. Won't the melting temperature keep dropping with the concentration?
20. The term "depletion length" is used, with any definition (I think).
21. Lines 308/309: R_g is used twice when I think $\langle r^2 \rangle$ is meant at least once.
22. The fit in Fig 5: it's worth emphasizing that quite a lot of curves could go through points with those error bars. I'd also like the parameters that were fitted to be explicit in the main text, rather than just SI, and it would be good to give some slightly more concrete evidence about why they are consistent with known geometry.
23. Start of caption of Fig 5: I'd say "Rings formed from DNA nanotubes..." rather than "DNA nanotubes...".
24. The CG model definition, and the ring diameter metric, that are defined in the SI should be referenced more explicitly from the simulation methods section.
25. Fig 3 of SI. Is it possible to say what the two rows are in the caption? Is it also possible to increase the contrast in the lower row?
26. Supplementary Note 1 contains quite a bit of text repeated word-for-word from main text.
27. When referring to figures in the SI, I think it would be best to use "Supplementary Figure XX" rather than "Figure XX".
28. I was a bit confused about the discussion of "reduced bending stiffness" in the SI. As I understand, the CG models are designed to quantitatively predict phenomena anyway (are the other parameters tightly controlled?) I was also confused by what I'm supposed to take from the reference to Fig 3e at this point.

29. As I understand the argument about penetrable shells, the story is that the crowders can penetrate, but the rods do not enter within the shell when binding (thereby reducing the effectiveness of small crowders). Is this the essence of the effect? It took me a while to reach this understanding, it might be helpful to clarify.

30. Just above eq. 30 in the SI. It's unclear to me what is "fitted" for $D/\sigma \sim 100(e/kT)^{-2/5}$. Is it the "100"?

31. I don't think "data is available on request" is really good enough in this day and age. Shouldn't data and processing scripts be uploaded to a repository? It would also be good practice to share simulation code.

Point-to-point response

Reviewer #1

“ In their manuscript, Illig & al. present a novel DNA nanotechnology system consisting of DNA nanotubes that self-assemble from DNA tiles into micron-sized rings. The motivation and goal of this work are to achieve a division mechanism for systems such as synthetic cells, which is an invogue synthetic biology field of research. The results shown in this manuscript demonstrate that DNA nanotechnology could be a new route towards this goal, albeit it would require additional extensive work. The DNA nanotubes, in the presence of a multivalent positively charged block co-polymer (starPEG) made of a four-armed PEG core and positively charged short Lysine-Alanine peptide repeats, further self-assemble into bundles that can loop onto themselves and form rings. They first show that the formation of those rings is due to the multivalent electrostatic interactions between the DNA tiles self-assembled into nanotubes, and the positive charges on the four-armed starPEG cross-linkers. On the one hand, they show that the diameter of those rings is independent of the concentrations of DNA tiles - provided that a ten-fold molarity of starPEG is present - and of the magnesium ions. On the other hand, they show that this system is stable with time. They provide a theoretical frame that describes the formation and the time evolution of the DNA rings by considering them as a single DNA nanotube coiled multiple times around itself, that traps a gas of crosslinkers in its interior. The model takes into account the bending rigidity of the coiled DNA nanotube, an adhesion force between the different coils of the DNA nanotube in the interior of the DNA ring, and the free energy of the gas of crosslinkers. Using this model, theoretical predictions are made with respect to the dependence of the diameter of the DNA ring on the bending stiffness of the DNA coil and the adhesion force between the coils. The predictions about the bending stiffness of the DNA coil are tested experimentally by changing the temperature of the system, while the ones related to the adhesion force are tested experimentally by adding a depletion-attraction force mediated by a molecular crowder consisting of Dextran of increasingly large molecular weights. They successfully report the decrease in the diameter of those DNA rings as they increase the temperature of the system or add molecular crowders, consistent with the predictions of their model. Overall, the work shows an interesting first step towards achieving a division mechanism based on DNA nanotechnology. The manuscript is clear and concise for the most part. The experiments are well described and illustrated by clear figures and provide a convincing characterization of the DNA rings. The conclusions and perspectives about the extent of the work are honest. ”

Reply: We thank the reviewer for the interest in our study, for the positive words and for the valuable comments.

Comment 1: *“The description of the model is not as clear. A figure would be very helpful to give a visual representation of the different parameters that describe the DNA ring and the interactions that take place.”*

Reply: We thank the reviewer and agree with this remark. We have prepared a new Figure 4, copied below (Response Figure 1) and integrated into the main text of the revised manuscript, highlighting all the parameters of the model in subfigures **a** and **b** to improve the clarity of our investigated system.

Comment 2: *“Although a mapping of the lengths between simulations and experiments is provided, none is given for the time. It would be interesting to give the simulation time Δt in actual time and compare the dynamics obtained from simulations to those observed in experiments. The duration of annealing used in MD simulations could, for example, also be compared to the one used in the synthesis of the nanotubes.”*

Reply: We thank the Reviewer for insisting on this point. In MD simulations we employed an unphysically low viscosity (a factor 10^5 smaller than the viscosity of water) to accelerate equilibration. For such low viscosities we obtain a total equilibration time of only ~ 20 ms in MD simulations. From this low viscosity results, we need to estimate an equilibration time at realistic higher viscosities.

In the overdamped limit of Brownian dynamics at high viscosities, the time scales for a given physical process such as equilibration by ring contraction scale strictly linear with viscosity because all processes are slowed down proportional to friction. In the underdamped limit at low viscosities, however, the time scale becomes essentially independent of viscosity. The boundary between overdamped and underdamped regime is where the Lennard-Jones times scale equals the frictional time scale. This argument enables us to obtain the time scale for equilibration at realistic viscosities from our low viscosity MD simulation. We predict an equilibration time of ~ 20 s, which is too fast to be followed directly in experiments (essentially, mixing DNA nanotubes with crosslinkers and filling the solution into an observation chamber takes at least half a minute before the microscopy experiment can be started).

To further support our arguments, we also present MD simulations of a faster process, namely the pulled sliding of only two polymers, where we can actually explore a wide range of viscosities in MD simulations and measure the time to desorption for a range of viscosities. These results fully confirm the proposed scaling with viscosity.

We present the scaling argument for viscosity changes as well as the additional supporting simulations in a new subsection “Time scales of MD simulation” of the Supplementary Note 2. The estimates of equilibration time scales for ring contraction are mentioned in the main text in the section on MD simulations.

We also added a paragraph to the main text (ll. 288-294): “Investigations on a translation of the low viscosities applied in the MD simulation to the more realistic viscosity of water predict an equilibration time of ~ 20 s (see Supplementary Note 2 for details). These results indicate that ring contraction happens too fast to be observed in our experimental setting because it takes at least half a minute before the microscopy experiment can be started due to mixing of DNA nanotubes with crosslinkers and crowders and filling the solution into an observation chamber.

Response Figure 1: (New Main Fig. 4) Theoretical description and simulation of DNA nanotube rings. **a**, Sketch of the theory model illustrating the associated parameters. The DNA nanotubes are drawn as a continuous line and colored to distinguish surface (red) and interior (yellow) of the ring. The diameter of the bundle cross section is assumed to be negligible compared to the ring diameter D . Zoom: A discretized bead-spring representation of the DNA nanotubes is used in the MD simulations with parameters as indicated. **b**, Snapshots of an isotropic initialization (left) and a DNA nanotube ring (right) taken from MD simulations. For clarity, the filament widths are increased. The cubic boxes show the simulation volume. **c**, Coarse-grained MD simulation of the DNA ring formation from a solution of DNA nanotubes represented as bead-spring polymers. Individual nanotubes involved in ring formation are colored for clarity. **d**, Kinetically trapped structure in incomplete ring formation after simulated annealing (right) (starting at temperature T_1 (left) and annealing to a high temperature $T_2 = 8T_1$). **e**, Electron microscopy image of a kinetically trapped DNA ring as observed in experiments. Scale bar: 500 nm.

In addition, we can only analyse DNA rings immobilized at the surface which prevents imaging during ring contraction.”

Comment 3: “ *A brief comment about the choice of T_2/T_1 with respect to ϵ would be useful.*”

Reply: To underline our choice of T_2/T_1 , we added the following paragraph to the Supplementary Information, Supplementary Note 2, MD simulations, Equilibration: “The value of T_2 is adapted depending on the simulated potential strength ε to prevent complete unbundling of the ring structure during the interval at T_2 . For small potentials $0.2 < \varepsilon \leq 0.5 k_B T$, an annealing temperature $T_2 = 2T_1$ is sufficient to equilibrate the rings within achievable simulation times. The annealing temperature is gradually increased with the potential up to $T_2 = 8T_1$ for $\varepsilon = 2.0 k_B T$ because annealing at $T_2 = 2T_1$ turns out to be insufficient to “loosen” the bundle effectively for these larger values of ε . For very small $\varepsilon/k_B T_1 \leq 0.2$, we do not use simulated annealing.”

Comment 4: “*In Figure 4, for the larger value of potential strength (yellow crosses), why is there a step of the value of the diameter at lower values of the stiffness? That deviation from the provided scaling law should be commented on.*”

Reply: We thank the Reviewer for pointing this out. The three simulations corresponding to the first three yellow crosses in Fig. 5 correspond to the largest values of attraction and smallest bending rigidity resulting in the smallest diameter. In this regime, equilibration is difficult because of the lock-in of neighboring polymers. The lowest three curves in the Response Fig. 2 show the resulting bundle contraction during equilibration, which feature rather slow fluctuations around the equilibrium. This hints at large autocorrelation times. Ring diameters were measured in a fixed time interval between $8-10 \times 10^8$ MD time steps. Therefore, deviations of these data points from slow fluctuations are larger and the apparent “steps” compatible with the scaling law.

Response Figure 2: Bundle contraction during equilibration (ring diameter as a function of MD simulation time) for different single DNA nanotube rigidities κ for $\varepsilon/k_B T = 1.0$.

As suggested by the Reviewer we included a comment about these details of the equilibrium ring diameter measurement and the Response Figure 2 (as new Supporting Figure 14b) in the Supplementary Note 2: MD Simulations in the section: “Equilibration”.

Comment 5: “ *Some typos: Line 131, blank space missing before “StraPEG” Figure 4e) caption, blank space is missing in “decreasingpotential” Line 309, $R_g \ll d$ to replace with $R_g \gg d$ Line 450, a period is missing before ‘The pre-processing’* ”

Reply: We thank the reviewer for pointing out the typos. We fixed them accordingly. Regarding line 309, we agree with the reviewer, that the correct notation should be $R_g \gg d$.

Reviewer #2

“ This is a nice paper reporting novel and potentially useful information on the synthetic contractile rings. In this paper, the authors summarize important aspects related to the formation of micrometer-scale rings using DNA nanotubes and their contraction, which are crucial for creating artificial cells. The main results they presented are as follows: - Authors demonstrated the formation of micrometer-scale DNA nanotube rings that are crosslinked by starPEG-peptides. - The bundling of DNA nanotubes by starPEG-peptides is due to crosslinking through their charge, not depletion force. - Based on the theory that crosslinkers maximize the overlap region between filaments due to entropic force, authors described the parameters and characteristics of the DNA rings. - Authors reproduced the contraction process of the ring based on the theory using CG-MD simulation. - Furthermore, authors confirmed the contraction of the ring by temperature changes and addition of molecular crowders, based on the prediction from the theory. Overall, it is novel and surprising that micrometer-scale rings can be formed using DNA nanotubes and that they are contracted to less than half of their original size. The process of verifying the characteristics predicted by theory and MD through experiment is impressive. However, I believe some concerns should be addressed before publication. ”

Reply: We thank the reviewer for the kind words regarding the novelty and interest in our results.

Comment 1: “ *The only major criticism I have is that the advantages of using DNA nanostructures are not maximized, and there seems to be a lack of description or experimentation. Why use DNA instead of naturally-occurring actin filaments? The advantages of using DNA nanotubes include controllable stiffness, easy functionalization, sequence addressability, and application of powerful genetic tools. If such experiments are not feasible, it would be helpful to elaborate further on the advantages of using DNA nanotubes in artificial cells for bottom-up synthetic biology. I understand the power of DNA nanotechnology and that references such as 12–16 and 20–24 are already listed in the introduction section. However, most readers may not have a clear picture, so highlighting the specific advantages of using DNA nanotubes over cytoskeletal filaments for future artificial cells would help readers understand the importance of this paper. ”*

Reply: The reviewer's comment is legitimate and we thank the reviewer for giving us the opportunity to elaborate further on the advantages of using DNA nanotechnology.

In the following, we want to (I) briefly comment on DNA design potential as an entirely synthetic material, (II) point out the advantages of DNA nanotubes we applied in this study and (III) discuss their advantages over actin filaments for bottom-up synthetic biology.

(I) We chose the DNA nanotubes as a well-established and adaptable system, which has already been used to mimic the cytoskeleton – yet without achieving contractile rings. It is exciting to build a protein-like machinery from non-protein based parts from a fundamental point of view. We did not alter geometrical or chemical properties by DNA design because it has been done in the past by us and other groups (as cited in the main text). In addition, geometry changes like stiffness by design (Schiffels et al. 2013, ACS Nano, DOI:10.1021/nn401362p) or photocontrollable stiffness (Sethi et al. 2023, Nanoscale, DOI:10.1039/D2NR05202D) have been achieved.

To underline this, we edited the main text (ll. 391-394): “Since our micron-sized DNA rings consist of well-established DNA nanotubes, further investigations, e.g. involving mutations like stiffness control (Schiffels et al., Sethi et al.) are facilitated compared to protein-based materials like actin filaments.”

(II) We agree with the reviewer that DNA nanotechnology offers many more advantages (some of them listed above). Although it is not the main focus of our study, we already make use of one of them: Temperature change affects DNA hybridization and thereby also melting ends of DNA nanotubes, which has also been confirmed by simulations (Markegard et al., 2016, The Journal of Physical Chemistry, DOI:10.1021/acs.jpcc.6b03937). It also affects the persistence length of DNA as cited in the main text, and we describe what role temperature plays for the ring contraction. Therefore, we actually do make use of the temperature dependent base-pairing of DNA.

DNA nanotechnology allows us in general to skip the step of purification in our experimental protocols compared to proteins. Actin filaments would require cell culture, protein purification, delicate buffers and result in experimental hurdles for an upscale of the production yield. DNA is inexpensive and easy to handle.

To underline this, we edited the main text (ll. 395-399): “In addition, synthetic DNA rings are already equipped with features responding to temperature change similar to their biological counterparts (polymerization of actin filaments (Zimmerle et al. and DNA nanotubes (Hariadi et al., Winfree et al.)). By contrast, DNA nanotubes can easily be equipped with other types of molecular functionalization, reprogrammed and repurposed for diverse systems.”

(III) One can argue on a conceptual level whether synthetic cells should be build up of protein-based materials or an entirely bottom-up engineered machinery built from DNA or other materials. We believe that both, protein-based syncells and DNA-based syncells are exciting. The key aspect, that led us to use DNA is the fact that DNA can, in principle, be copied more easily. Nucleic acids can be self-replicated with a much lower amount of components compared to proteins (which, according to the central dogma need the entire transcription-translation

machinery consisting of approximately 150 components).

To follow the reviewer's suggestion to provide the reader with the according outlook and highlight the specific advantages more clearly, we added the following passage to the main text (II.57-63):

“Here, we revert to the DNA nanotubes as an entirely synthetic system that is well established in the bottom-up synthetic biology community as an alternative route to the reconstitution of proteins. The reconstruction of protein-like machinery from a different material is not only exciting in itself, it may also provide a shortcut towards a truly self-replicating system since DNA replication required fewer components than the replication of proteins. In the long term, one could thus envision a synthetic cell which operates outside of the central dogma of molecular biology.”

Comment 2: “ *Minor points: - Line 16 in Abstract: I found it somewhat difficult to understand what the authors refer to by “these two conditions.” It may be clearer if the clause in the previous sentence is changed to, for example, “–upon (i) increasing attraction or (ii) decreasing bending rigidity of the DNA nanotubes, . . . ”* ”

Reply: In the abstract we write that theory and simulations predict ring contraction upon increasing attraction or decreasing bending rigidity. It is in fact not as simple as calling out “these two conditions”, since especially increasing attraction can be realized in various ways, e.g. by temperature or depletion effects. In the theory review we introduce a number of scenarios (i) to (iv) which can be tested experimentally and discuss their feasibility. In summary two parameters L_p and g can be controlled in simulations and experiment. As we want to keep the detailed description of scenarios (i)-(iv) we cannot double use the nomenclature but are happy to change the text in the abstract to (II. 16-17): “We experimentally realize a variation of these parameters by addition of molecular crowders or temperature increase, respectively.”

Comment 3: “ *- The length distribution of DNA nanotubes seems to be missing. Does the initial length of DNA nanotubes affect the formation and contraction of the ring? How the experimentally obtained length of nanotubes is implemented in the simulation? ”* ”

Reply: The length distribution of DNA nanotubes follows a Poisson distribution and can be found in two of our earlier publications from which we reproduced the DNA nanotube formation protocol (Figure 1c in DOI: 10.1021/acsnano.1c10703 and Figure 2b in DOI: 10.1038/s41557-022-00945-w). Our results already show that we can form rings from DNA nanotubes with different lengths (Fig. 3a and Supporting Fig. 4). We believe that ring formation requires sufficiently long DNA nanotubes that can overlap within one ring. Therefore, the bending stiffness of DNA nanotubes provides a lower limit for the formation of small rings. The smallest observed rings had a diameter of 500 nm.

For the simulations, the experimentally obtained DNA nanotube length distribution has been

implemented by modifying the number of beads per nanotube. The nanotube length is drawn from a Poisson distribution with the experimentally obtained mean value of $580\sigma \approx 6.96 \mu\text{m}$.

We edited the main text (ll. 252-253): "The lengths of the simulated DNA nanotubes are Poisson distributed, as experiments from previous publications revealed, and set to $580\sigma \approx 6.96 \mu\text{m}$."

Comment 4: " - Fig. 3b: The description "Representative confocal (left) and STED (right) images" should be replaced with "Representative confocal (top) and STED (bottom) images" ? "

Reply: We thank the reviewer for this remark and edited the Figure caption of Fig. 3b accordingly: "Representative confocal (top) and STED (bottom) images".

Comment 5: " - Fig. 5d: Is this the result of MD simulation? not experiment? "

Reply: Fig. 6d depicts the experimental data from Fig. 6c to the depletion theory. The origin of the black data points was indeed not specified. Therefore, we edited the figure caption as follows: "Experimental DNA nanotube ring diameter reduction by depletion attraction as a function of molecular weight of dextran at 25wt% (black, identical to data points in c, constant total number of dextran monomers)."

Comment 6: " - Line 205: There is no "iv.b", so "iv.a" is not required? "

Reply: As a result of the theoretical review, the scenarios (i) - (iv) were derived. We distinguish two special conditions in (iii), namely (iii.a, iii.b) and only one special condition in (iv), namely (iv.a). The latter is referenced to later on in the main text and is a special case of (iv). This is why we chose to keep the notation as it is.

Comment 7: " - Line 165: The technical term 'gas' may be a common term among theoretical people, but I feel that it would be helpful to provide an explanation for ease of reading. I should have read some previous papers to find out what exactly 'gas' means. "

Reply: This term was used and introduced in Ref.[4] (Kucera et al. Anillin propels myosin-independent constriction of actin rings. Nature Communications 12, 1–12 (2021)). This tempted us to use it without further explanation. We agree with the Reviewer that it is better to explain the term properly when it is introduced.

For clarity, we added the following explanation to the main text (ll. 196-199): "Crosslinkers generate an adhesive energy between two DNA nanotubes by presenting one adhesive end to each DNA nanotube (Kierfeld et al.). Thereby they accumulate in the overlap region between DNA nanotubes and can be viewed as a one-dimensional gas of particles confined to the overlap region. "

Comment 8: “ - Line 347: I was wondering if the reference number is correct: “Our approach could be complemented by engineered molecular motors that walk on DNA nanotubes[37].” ”

Reply: Thank you. Indeed the reference was wrong and we replaced it with Ibusuki et al. (Science 2022, DOI:10.1126/science.abj5170).

Reviewer #3

“ Illig et al. present an experimental study, supported by basic theory and coarse-grained simulation, of contraction of synthetic DNA rings. The system is intended to mimic contraction of cytoskeletal rings, which play an important role in processes such as cell division. The authors demonstrate that they can achieve impressive degrees of contraction by either adding molecular crowders or temperature increase. Qualitatively, this behaviour is consistent with the predictions of the simulation and the basic theory. Overall, the system is interesting and the evidence persuasive. The manuscript is also well written. I am therefore inclined to recommend publication, subject to addressing the technical criticisms below. Please note that I am not an expert in the experimental methods here and I am not in a good position to comment on their reliability. The approaches seem reasonable to me, and plenty of detail is given. ”

Reply: We are pleased about the reviewer’s recommendation to publish our manuscript and thank for the detailed comments, which we addressed below.

Comment 1: “ *It wasn’t that clear to me what the authors want the overall interplay between the theory, simulation and experiment to be. Is the idea that the simulations test whether the basic theory is reasonable; having done so, this basic theory is then used to select certain triggers for forcing the system to contract? The authors don’t quite build enough of a narrative to hold these three parts together. Perhaps simply stating more explicitly how each part of the work relates to the overall goal would help. For example, how does fitting the scaling of the simulation data to the theory in 4e help with the overall goal? ”*

Reply: We thank the reviewer for this comment, which helped us to improve the flow of the text. To be fully transparent, we first set out to construct a contractile machinery from DNA experimentally. We tried different strategies, including hairpin loops for strand displacement, without success. We then decided to go for a more biomimetic strategy using crosslinkers and we were at first surprised about the immediate success. We then turned to theory to explain our observations. So to be clear, experiments were first.

In the text, we first reason the use of DNA nanotubes and the peptide construct for the

self-assembly of micron-scale rings which turns out to be evident from several theoretical perspectives. Next, we develop a theoretical review that enables us to understand the mechanism at play from physical parameters, e.g. attraction strength, which we discuss in detail. The extracted parameters can be translated one-to-one to the coarse grained MD model and the mechanism of contraction can be tested in simulation. Our original physical system of interest, the DNA nanotube rings, show a constriction upon parameter change that are relatable to the previous theoretical results. However, those initially determined success parameters can not be translated one-to-one to the experiment by definition. Several dependencies especially the temperature dependence interlink many features of the complex system. The attraction strength itself for example cannot be addressed without the parallel manipulation of several system parameters as it is the case for molecular crowding.

To clarify this, we edited the narrative in the introduction (ll. 73-79): “With theory and molecular dynamics simulations we gain mechanistic insights into the formation of DNA nanotube rings and the architecture of its contraction mechanism. We translate the simulation parameters of interest into physical properties of our system and realize the predicted conditions experimentally. Thereby, we achieve the contraction of the DNA rings to less than 45 percent of their initial diameter. We relate this to the theory and adapt it to the particularities of the physical system so that we can reduce the entangled relationships of the experiment to quantitative parameters.”

In addition, we added the following sentence introducing the experimental results (ll. 296-297):

“Finally, we set out to realize DNA ring contraction experimentally and use simulations and theory to rationalize the results quantitatively.”

Comment 2: “ *A lot of the figures are very small. Is this really necessary?* ”

Reply: We followed the guidelines for figures to be either single column or double column wide. Recommended font sizes of Nature Communications are between 5 and 8pt and we chose to be in the larger half range with 7pt and ensured readability. We also chose the data contents in the subfigures to be equally large (compare to Fig. 5a,b,c) or comparative data to be captures at once (Fig. 3c,d) and thereby maintaining a compact appearance. The adapted additional Figure 4 shortens the remaining Figures 3 and 5. We are happy to make further changes to the figures if the editor suggests so.

Comment 3: “ *In some cases it isn't immediately obvious what the reader should take from the figures. Eg. Fig 2 - what do I learn from 2b? What is it about the figure that tells me this?* ”

Reply: Figure 2 focuses on the characterization of the DNA nanotube bundle crosslinked by the peptide construct. Here we first show in a qualitative way how two DNA nanotubes are crosslinked by starPEG-(KA7)₄. To proof this, we prepare two DNA nanotubes with different fluorophores and fix the first one, apply washing steps and then observe the second one to colocalize in a composite confocal image (Figure 2b). Colocalization here means the bundling of

two individuals, fixed nanotube A and free nanotube B by crosslinker gas. We have supplemented the figure caption as follows: "...illustrating their colocalization".

Comment 4: *"I'm unclear on the purpose of the exponential fit in 2d and 2f. Is there a theory that says an exponential is expected? Otherwise I would not recommend fitting (although a spline to guide the eye may be reasonable)."*

Reply: We expect both, the association of starPEG-(KA7)₄ and the association of DNA nanotubes to bundles, to follow association binding kinetic models. Therefore, it can be fitted with an equation of the form: $y = y_0(1 - \exp(-kx))$ with x being the starPEG-(KA7)₄ concentration.

Comment 5: *"In the discussion of the ring geometry, I was initially very confused because I was expecting some discussion of what drives the ring to form in the first place. In fact, I think the authors start from the assumption that rings have formed and then discuss how the geometry changes in response to the environment. It would be helpful to make this clear up front."*

Reply: Indeed we observe the self-assembly of rings formed from DNA nanotube bundles and have no experimental observation of their formation. However, in the simulation we learn how the ring formation could eventually happen. In general, we assume DNA nanotubes with a persistence length of around its mean length (the lengths of DNA nanotubes follow a Poisson distribution). The persistence length is defined as the length over which correlations in the direction of the tangent are lost. This lets us assume that within this parameter space an encountering of DNA nanotube ends is possible due to the statistical movement of its ends. The crosslinker helps to maintain overlaps and induces the bundle to grow.

In order to clarify that we already gain insights into ring formation with the CG-MD simulations, we added the following sentence to the introduction (ll. 73-75): "With theory and molecular dynamics simulations we gain mechanistic insights into the formation of DNA nanotube rings and the architecture of its contraction mechanism."

To clarify, we have included the following paragraph in the main text, which refers to the order of experiments we have made (ll. 120-126): "The extent of bundling correlates with the concentration of starPEG-(KA7)₄ relative to the concentration of DNA tiles (Supplementary Fig. 1), forming a hybrid material with engineerable properties. The bundles curve into closed rings. To tune ring sizes and accomplish contraction, we first need to understand the action of the peptide crosslinker on DNA nanotubes. In particular, the bundle formation is discussed in the following paragraphs. The characteristics of the ring formation and ring structures are considered after that."

In the section 'Self-assembly of micron-scale DNA rings' we have supplemented the following paragraph to determine our starting point for further consideration (ll. 169-177): "Since the persistence length to mean length ratio for DNA nanotubes (Poisson distributed lengths) is around 1 and the persistence length is defined as the length over which correlations in the direction of the tangent are lost, we can assume that an encountering of DNA nanotube ends

is likely. Once the ends have met, some DNA nanotubes may undergo end-to-end joining, others overlap to maximize the crosslinking. STED reveals free DNA nanotube ends at the edges of the ring suggesting that the observed ring formation cannot be a result of end-to-end joining only, but is rather mediated by crosslinkers along the DNA nanotubes that induce further growth of the bundle thickness by recruiting more single or prebundled DNA nanotubes (Fig. 3b, Supplementary Fig. 7).”

To allow the reader to be guided more clearly from this section to the following section on molecular simulations, we also edited the summarizing paragraph (ll. 185-189):

“This ring formation mechanism seems to mimic naturally occurring ring formation by cytoskeletal filament crosslinking. To gain a deeper understanding for DNA nanotube ring formation and to derive strategies for potential ring contraction, we next develop a theoretical framework and subsequently combine it with coarse grained molecular dynamics (MD) simulations.”

Comment 6: “ *The gas gives an ”additional” effective adhesion. Additional to what? Are the authors saying that the crosslinker have two contributions, one of which is this entropic term?* ”

Reply: The reviewer is exactly right. Each crosslinkers gives an adhesion enthalpy from its sticky ends. In addition the one-dimensional crosslinker gas in the overlap region between DNA nanotubes also has an entropy, which is maximized if the available “space” (i.e., the overlap region) is maximized. This corresponds to an entropic contribution to the free energy of DNA nanotube adhesion that wants to maximized the overlap. This is exactly the 1D Tonks gas contribution $F_c(L_i)$ in eq.(2).

We added an additional explanation of the notion of a crosslinker ‘gas’ (see also comment 7 of Reviewer 2), which should also further clarify this point.

Comment 7: “ *I would recommend having a figure in the main text to illustrate the theoretical concepts discussed on p9-11, and the features of the CG model, schematically. It would clarify a lot of what is currently discussed only in words, with terms like ”overlap” that are not formally defined. ”* ”

Reply: To clarify the theoretical model and its parameters as well as the representation in the simulations, we have partly reorganized the respective figures and included a sketch visualizing the theoretical model and the underlying ideas (Fig. 4a).

Comment 8: “ *What do the colours mean in Fig. 3e? ”* ”

Reply: The colors are randomly chosen and depict individual simulated DNA nanotubes. In the background some DNA nanotubes that are not involved in the current ring formation are still yellow. The coloring helps to guide the eye to find the endings of partly bound DNA nanotubes.

We edited the Figure caption accordingly: “Individual nanotubes involved in ring formation are colored for better visibility.”

Comment 9: “ *I think it’s worth saying that the model assumes uniform bending in a filament of uniform susceptibility.* ”

Reply: We added further specification of our toroidal ring model before eq.(2). We added two sentences (ll. 204-205; 208-209): “We consider a torus of diameter D consisting of bundled DNA nanotubes, which are uniformly bent.” and “We assume uniform DNA nanotube bending rigidities throughout the entire torus.” We hope that this clarifies the issue raised by the Reviewer.

Comment 10: “ *I initially thought D was the diameter of the cross-section of the bundle; it may help to be more explicit.* ”

Reply: This is a valid point. We edited the initial definition of D as follows (ll. 204-206): “We consider a torus of diameter D consisting of bundled DNA nanotubes. The bundle’s circular cross section contains N DNA nanotubes resulting in a total length $L_{\text{tot}} = \pi ND$.” We further introduce all parameters in a new figure (Fig. 4a) to avoid such confusion.

Comment 11: “ *What does b_c describe in the actual physical system? Is it the size of the cross-linker’s footprint?* ”

Reply: We introduce b_c as the “size” of the crosslinker, which we use to quantify the minimal distance between crosslinkers as we state in the main text “with a line density of $1/b_c$ of possible binding sites” just following eq.(2). In this sense the Reviewer is exactly right to view it as the “footprint” of the crosslinker.

We find the specification of a “line density of binding sites” more precise than the notion of a “footprint” of the crosslinker but this is a matter of taste. We added the sentence “In this sense b_c can be viewed as the size of the “footprint” of the crosslinker on the DNA nanotube.” as the idea of a “footprint” might actually be a very useful picture to have in mind (ll. 219-220).

Comment 12: “ *The derivations performed using the model (reported in SI) should be explicitly referenced in the text. For example, where is Eq. (3) derived?* ”

Reply: We did exactly that in referring in the sentence including Eq.(2) and the following sentence twice to Supplementary Note 1. We tried to improve the presentation by adding one additional reference within the sentence including Eq.(3) and by including references to the corresponding equation numbers in the Supplementary Note 2.

Comment 13: “ *I found the $2+\alpha$ notation slightly ambiguous. I don’t think 2 and alpha are supposed to be in the same denominator, but it might be misinterpreted that way.* ”

Reply: We used the standard notation that is used, e.g., by all APS journals: $3/2 + \alpha = (3/2) + \alpha$. We added brackets for complete clarity both in Eq.(3) and in Eq.(9) in the Supplementary Note 1.

Comment 14: “ *The actual definition of the CG model is not very explicit in/explicitly referenced from the main text. The schematic diagram would help here, as it would help to explain the attraction to ”two neighbouring beads”, which I don’t think is hard-coded but arises from the way the chains fit together. ”* ”

Reply: We added Fig. 4a containing such a scheme (see also reply to comment 7).

Comment 15: “ *Temperature ranges of 2-8 T_1 are not ”comparable to typical experimental” protocols, where in the temperature typically changes by 25% or less. ”* ”

Reply: The Reviewer is completely right, and we do not claim in the text that the annealing procedure used in simulations is related or can be compared to any experimental procedure. We merely do remark that, occasionally, we observe both in experiments and during the annealing procedure strikingly similar unbundled trapped structures (Fig. 4). This is a hint that kinetic trapping might occur both in simulation and experiment (caused by different non-comparable protocols).

We added a remark to the main text (ll. 266-267): “(we note that experimental and simulation protocols giving rise to trapped structures can not be compared due to the different procedures)”.

Comment 16: “ *Fig 4a isn’t well described by the caption. What are all the subfigs showing? I think only the third subfig is ”trapped”. ”* ”

Reply: The Reviewer is correct. The caption of Fig. 4d (former Fig. 4a) has been improved.

Comment 17: “ *Given the importance of temperature scaling, it would be helpful to state which parameters are assumed to be temperature-independent (eg. Kappa) when they are introduced. ”* ”

Reply: Most of the parameters of a biological system are temperature-dependent. The persistence length of DNA is indeed temperature dependent by definition, $L_p = \kappa/k_B T$, while the bending rigidity κ itself should be largely temperature-independent. This can be translated to the 3-dimensional assembly of dsDNA, so we conclude the same for DNA nanotubes. The bending rigidity κ is assumed to be temperature-independent in our model. Also L_{tot} is simplified to be T-independent, since we neglect the possibility of melting. Even the number of DNA nanotubes can be T-dependent due to entire melting of short DNA nanotubes.

We agree with the reviewer that it would be helpful to easily see which parameters are set to

be T-independent, although it frequently needs a rather detailed case distinction. We fear that specifying the temperature dependence (or independence) for each parameter in the main text would impede the flow of reading. And we thereby suggest an additional Supplementary Note with detailed discussion of the T-dependencies of the theoretical and simulated parameters and their translation to the physical system (see Supplementary Note 3).

Comment 18: “ *Top of page 16: paragraph starts with ”to compare”, but I don’t think a comparison follows. ”* ”

Reply: We appreciate the reviewer’s comment and edited the wording to “To relate” (l. 305)).

Comment 19: “ *On p16, the authors talk about a melting temperature ”below 5nM” of free tile. Won’t the melting temperature keep dropping with the concentration? ”* ”

Reply: We agree with the reviewer and edited the wording to “Our DNA nanotube design has a maximum melting temperature of 37.2 °C (at a maximum free tile concentration of 5 nM, compared to 42.0 °C at 50 nM free tiles).” (ll. 311-313).

Comment 20: “ *The term ”depletion length” is used, without any definition (I think). ”* ”

Reply: We thank the Reviewer for pointing out that the definition was not very clear. The depletion length δ was introduced as “depletion layer thickness” without introducing the term “depletion length” right away. We changed this in the new version of the manuscript (ll. 363-366).

We also added a more concise definition in the Supplementary Note 3 for completeness: “The depletion length δ is defined as the depletion layer thickness via $\delta = \int_0^\infty dx(1 - f(x))$, where x is a coordinate measuring the distance from a flat surface and $f(x) \in [0, 1]$ the relative polymer segment distribution. ”

Comment 21: “ *Lines 308/309: $Rg; d$ is used twice wen I think $\zeta; \zeta$ is meant at least once. ”* ”

Reply: We agree with the reviewer, that the correct notation should be first $Rg \ll d$ and second $Rg \gg d$ (l. 355; l. 356).

Comment 22: “ *The fit in Fig 5: it’s worth emphasizing that quite a lot of curves could go through points with those error bars. I’d also like the parameters that were fitted to be explicit in the main text, rather than just SI, and it would be good to give some slightly more concrete evidence about why they are consistent with known geometry. ”* ”

Reply: The large uncertainty arises from the experimental setting. Sources of uncertainty are for example pipetting errors for dextran solutions of high viscosity. Also mixing at high viscosities might incorporate air into the sample. Imaged and analysed rings are very heterogenous just like DNA nanotubes lengths, too. Their length follows a Poisson-distribution as one may expect which leads to highly heterogeneous bundles and also ring sizes. A comment on that is added to the

main text “The high uncertainty of ring diameters might mainly result from highly heterogeneous bundles that consist of DNA nanotubes of different lengths (Poisson-distributed).” (ll. 339-341). Although, when considering the average ring sizes the curve described the experimental data in fact very well.

The fit parameters were actually explicitly contained in the main text, directly in the caption of the Figure containing the fit. We repeat these numbers in the main text explicitly in the new version of the manuscript.

We agree with the Reviewer that we were too short on the discussion of these fit parameters and include a discussion of and comparison with the available experimental information on DNA nanotube diameters and crosslinker sizes.

Comment 23: “ *Start of caption of Fig 5: I’d say ”Rings formed from DNA nanotubes...” rather than ”DNA nanotubes...”.* ”

Reply: This is a legitimate point and we edited the Figure caption accordingly: “Rings formed from DNA nanotubes contract upon addition of a molecular crowder or heating.”

Comment 24: “ *The CG model definition, and the ring diameter metric, that are defined in the SI should be referenced more explicitly from the simulation methods section.* ”

Reply: As suggested by the Reviewer (and also in response to Reviewer 1), we inserted a reference at the end of the simulation methods section.

Comment 25: “ *Fig 3 of SI. Is it possible to say what the two rows are in the caption? Is it also possible to increase the contrast in the lower row?* ”

Reply: We increased the contrast for the images in the lower row in Supplementary Figure 3. The respective figure caption already contains the detailed information of what is depicted: the top row depicts DNA nanotubes (orange, labeled with Atto633) and the bottom row starPEG-(KA7)₄ (cyan, labeled with TAMRA). We edited the Figure caption as follows: “Colocalization of DNA nanotubes (top) and starPEG-(KA7)₄ (bottom).”

Comment 26: “ *Supplementary Note 1 contains quite a bit of text repeated word-for-word from main text.* ”

Reply: We designed the Supplementary Notes to be readable without having to look up too much additional information from the main text. We agree with the Reviewer that this leads to doubling certain information or even phrases from the main text. We do not see a serious problem here but are open for any advice from the editors.

Comment 27: “ *When referring to figures in the SI, I think it would be best to use ”Supple-*

mentary Figure XX” rather than “Figure XX”. ”

Reply: We thank the reviewer for pointing this out, we went through the text thoroughly and ensured the Figures are referenced correctly as the journal emphasizes.

Comment 28: *“ I was a bit confused about the discussion of ”reduced bending stiffness” in the SI. As I understand, the CG models are designed to quantitatively product phenomena anyway (are the other parameters tightly controlled?) I was also confused by what I’m supposed to take from the reference to Fig 3e at this point. ”*

Reply: We first want to point out that ring formation and ring contraction are different processes and are addressed in different simulation runs. They also have different computational problems. The challenge in ring contraction is to reach the proper equilibrium diameter after preparing an initial state with a ring structure. This is impeded by the problem that beads tend to “lock-in” to adjacent beads on neighboring polymers. The actual process of forming a ring (before it might contract by increasing attraction) is computationally challenging because of the time scales that are involved: ring formation takes a long time, which becomes prohibitively long for simulations if realistically stiff polymers are used (realistically stiff means $\kappa = 600k_B T \sigma$ for DNA nanotubes as stated in the paper) because the bending energy for ring closure will enter in an Arrhenius-type exponential law into the ring-closure rate.

Therefore, we employ a reduced stiffness of $200k_B T \sigma$ for the simulations, in which we aim to observe and follow the process of ring formation (for which snapshots are shown in Fig. 4c (former Fig. 3e)). We employ realistic bending stiffnesses of $600k_B T \sigma$ for all simulations aiming to equilibrate pre-assembled rings to their final equilibrium diameter (in Fig. 5 apart from Fig. 5a, where the effect of varying κ is explicitly studied).

We clarified this distinction of the two sets of simulations by adding a remark in the Supplementary Note and introducing a separate subsection “Ring formation”.

Comment 29: *“ As I understand the argument about penetrable shells, the story is that the crowders can penetrate, but the rods do not enter within the shell when binding (thereby reducing the effectiveness of small crowders). Is this the essence of the effect? It took me a while to reach this understanding, it might be helpful to clarify. ”*

Reply: The essence of the effect is that crowders can penetrate into the crosslinker shell of thickness p . This gives rise to a reduction of the depletion layer thickness $\delta \rightarrow \delta - p$ (because a length p of the crowder is “buried” in the penetrable layer). This shift gives rise to the changes in the depletion interaction, in particular, its dependence on molecular weight. Small low molecular weight crowders become less effective by this mechanism. In particular, very small crowders of size smaller than p will be completely buried in the penetrable layer and, therefore, not effective at all.

In order to clarify this, we added a corresponding explanatory sentence in the main text.

Comment 30: “ *Just above eq. 30 in the SI. It’s unclear to me what is ”fitted” for $D/\sigma \approx 100(\epsilon/k_B T)^{-2/5}$. Is it the ”100”?* ”

Reply: The Reviewer is completely right that the fit parameter is the prefactor 100 (assuming $\alpha = 1$ for sliding crosslinkers giving rise to the exponent $2/5$).

We clarified this by writing “Using the fit $D/\sigma = a(\epsilon/k_B T)^{-2/5}$ with a fit parameter $a \approx 100...$ ”

Comment 31: “ *I don’t think ”data is available on request” is really good enough in this day and age. Shouldn’t data and processing scripts be uploaded to a repository? It would also be good practice to share simulation code.* ”

Reply: We agree with the reviewer on the importance of data availability, we are happy to take care of this in the revision. We already provided a source data file with the first submission. The code is now available on GitLab <https://gitlab.tu-dortmund.de/cmt/kierfeld/triggered-contraction-of-dna-nanotube-rings> (the link will be replaced with a DOI to a zenodo archive as soon as the article is accepted), the data has been deposited on HeiData (<https://doi.org/10.11588/data/ADYUNN>) and can be accessed via this link after publication, a private URL that is currently accessible is sent to the editor as reference. We changed the data and code availability statement as follows: “The datasets generated during and analysed during the current study are available in the [Research Data] repository on heiDATA/Göpfrich Group - Biophysical Engineering of Life with the identifier <https://doi.org/10.11588/data/ADYUNN>. Source data are provided with this paper.”

REVIEWER COMMENTS

Reviewer #1 (Remarks to the Author):

The authors answered all the comments made in the previous review and improved the clarity of their manuscript accordingly. However, a point remains to be clarified: although the discussions around the mapping between the simulation time and the real-time (Comment 2) as well as the “step” observed in Fig. 5c of the new manuscript (Comment 4) are convincing, they both seem to hinge around the choice of the time interval for the measurements of the diameter of the ring.

In their response, the authors state that, on the one hand, they use an unrealistically low viscosity in their simulation to accelerate equilibration, and on the other hand, the “step” observed in Fig. 5c comes from slow fluctuations around the equilibrium that do not vanish by the time of measurement, which effectively increases the measured diameter of the ring.

Increasing the time of measurement would seem like a solution to both of the previous concerns. Is there a limit to that time of measurement? If yes, it should be commented on; if no, the answer to Comment 2 would be a more straightforward mapping of the time scales, and that to Comment 4 could be the display of the amplitude of the fluctuations in the form of error bars in Fig. 5c.

Reviewer #2 (Remarks to the Author):

The authors have appropriately addressed all my concern raised in the previous round of review. I recommend the publication in Nature Communications.

Reviewer #3 (Remarks to the Author):

The authors have responded in good faith to the comments, and their changes have improved the manuscript. I am basically happy for the paper to be published, with one significant outstanding point:

1. The authors say that the ring formation simulations are performed with different parameters from the contraction simulations. This is not clear from the main text or methods, as far as I can tell. Readers must be made aware of this fact without having to dig into the SI. Indeed, sometimes data is presented in the same figure which has been obtained with different models.

I also have a couple of optional comments that the authors may consider:

2. Related to the above point, the authors now say both that forcing a polymer of length $\sim L_p$ to loop is relatively likely, and that it's a very slow process, in different sections of the text. I agree with the latter - polymers of length L_p are hard to bend into a circle (although it is just about possible). It would be good to make these stories consistent.

3. Finally, several times in the response letter, the authors respond to my comments nicely, explaining the rationale behind things such as the exponential fit. However, it isn't always clear that the text has been edited to avoid other readers being similarly confused.

Point-to-point response

Reviewer #1

“ The authors answered all the comments made in the previous review and improved the clarity of their manuscript accordingly. However, a point remains to be clarified: although the discussions around the mapping between the simulation time and the real-time (Comment 2) as well as the “step” observed in Fig. 5c of the new manuscript (Comment 4) are convincing, they both seem to hinge around the choice of the time interval for the measurements of the diameter of the ring. In their response, the authors state that, on the one hand, they use an unrealistically low viscosity in their simulation to accelerate equilibration, and on the other hand, the “step” observed in Fig. 5c comes from slow fluctuations around the equilibrium that do not vanish by the time of measurement, which effectively increases the measured diameter of the ring. Increasing the time of measurement would seem like a solution to both of the previous concerns. Is there a limit to that time of measurement? If yes, it should be commented on; if no, the answer to Comment 2 would be a more straightforward mapping of the time scales, and that to Comment 4 could be the display of the amplitude of the fluctuations in the form of error bars in Fig. 5c.”

Reply: We thank the Reviewer for insisting on these points and are happy to elaborate further on our choice of the measurement interval and the motivation of the selected viscosity. We agree with the Reviewer on a conceptual level, that increasing the number of simulation time steps would render lowering of the viscosity unnecessary and would also allow to average out the observed fluctuations over an increased measurement time frame. However, the number of the required simulation time steps would increase by a factor of ~ 1000 according to our theoretical arguments as well as the results of the pulled sliding simulations (Supplementary Note 2), if we were to increase the viscosity to a realistic value. Given the computation time, i.e. wall time, of a typical simulation run of a few weeks, this is not a viable option.

To address the issue of the visible step in Fig. 5c further, we continued the simulations for the respective data points (at low value of the viscosity) to ensure that the simulations are fully equilibrated. We were able to double the simulation time, thereby shifting the measurement time frame to later simulation times $t > 1.8 \times 10^9 \Delta t$. This visibly reduces the size of the step (see new Fig. 5c in the main text) indicating that the simulations at low bending rigidities were actually not fully equilibrated (and are still not fully equilibrated after the longer simulation runs). For these three simulation points we can detect a small decreasing trend in ring diameter up to the longest simulation times $t = 2 \times 10^9 \Delta t$. In this sense, we have to revoke our previous statement where we attributed the deviation from the expected scaling behaviour to long autocorrelation times and, thus, insufficient statistics in an already equilibrated state. Nevertheless, the long autocorrelation time remains an issue for other simulation points. Both equilibration and autocorrelation times are considerably improved by the simulated annealing method. The simulated annealing procedure is less effective for the three data points with smallest ring diameter (smallest ratio κ/ε) in Fig. 5c because κ and, thus, the persistence length is smallest for these polymer rings. Annealing

effectively weakens the attraction by thermal shape fluctuations but small bending rigidities or persistence lengths reduce the resulting expanding forces caused by bending moments, which makes “loosening” of the bundle less effective. We changed the according passage in the Supplementary Note 2 “MD Simulations - Equilibration”.

In order to indicate that the questioned data points are actually not fully equilibrated, we marked them by a lighter yellow color in Fig. 5c and also shortly commented this additionally in the caption and the main text. Alternatively we could leave out these not fully equilibrated data points from Fig. 5c if the Reviewer finds this more appropriate. As these data points illustrate the limits of the MD simulation technique in combination with annealing, we opted to leave them in the plot but mark them with a lighter yellow.

Reviewer #2

“The authors have appropriately addressed all my concern raised in the previous round of review. I recommend the publication in Nature Communications.”

Reply: We are glad we could address all comments and grateful for the reviewer’s recommendation to publish our manuscript in *Nature Communications*.

Reviewer #3

“The authors have responded in good faith to the comments, and their changes have improved the manuscript. I am basically happy for the paper to be published, with one significant outstanding point:”

Reply: We are looking forward to address the reviewers comments.

Comment 1: *“The authors say that the ring formation simulations are performed with different parameters from the contraction simulations. This is not clear from the main text or methods, as far as I can tell. Readers must be made aware of this fact without having to dig into the SI. Indeed, sometimes data is presented in the same figure which has been obtained with different models.”*

Reply: We agree with the Reviewer that the reduced bending rigidity used for the ring formation simulations should have been mentioned in the main text. Accordingly, we complemented the caption of Fig. 4c with the sentence “A reduced persistence length is employed to facilitate the ring closure.” and added “The bending rigidity is reduced to $\langle L \rangle / L_p \approx 3$ in simulations of the ring formation to observe this process on computationally accessible time scales (see

Supplementary Note 2).” to the Methods section about the Molecular Dynamics Simulation.

Comment 2: *“I also have a couple of optional comments that the authors may consider: Related to the above point, the authors now say both that forcing a polymer of length L_p to loop is relatively likely, and that it’s a very slow process, in different sections of the text. I agree with the latter - polymers of length L_p are hard to bend into a circle (although it is just about possible). It would be good to make these stories consistent.”*

Reply: We thank the Reviewer for pointing out the inconsistent phrasing. The conclusion that ring formation is likely to occur was derived from the high yield of nanotube rings observed in the experiments (Fig. 3a) on the experimental time scale of several seconds to minutes. At the same time, we agree with the Reviewer that looping is a slow process on microscopic and simulation time scales. According to our simulations, this slow process becomes optimal at a ratio of $L/L_p \approx 3$ and, correspondingly, the experimentally realized lengths $L \sim L_p$ are close to this “sweet spot”. To avoid further confusion, the term “likely” has been replaced and the modified sentence reads “(...), we can assume that an encountering of DNA nanotube ends is possible on the experimentally relevant time scale of several seconds up to a few minutes.”.

Comment 3: *“ Finally, several times in the response letter, the authors respond to my comments nicely, explaining the rationale behind things such as the exponential fit. However, it isn’t always clear that the text has been edited to avoid other readers being similarly confused.”*

Reply: While answering all the Reviewer’s comments we have summarized responses and resulting text edits to the related topics and included the essential information in the main text. We have paid attention to not disturb the reader’s flow through the story and leave the paper concise. It may appear that the answer is more lengthy than changes in text. We would like to point out that long answers to specific questions are published alongside the paper.

To address the specifically mentioned point we fully agree with the reviewer and added the response regarding the exponential fit to the main text and included the fit parameters to the Figure caption as follows:

“We expect both, the association of starPEG-(KA7)₄ and the association of DNA nanotubes to bundles, to follow association binding kinetic models. Therefore, it can be fitted with an equation of the form: $y = y_0 + (P - y_0)(1 - \exp(-kx))$ with x being the starPEG-(KA7)₄ concentration.”

“**d**, Colocalization intensity ... exponential fit plotted as red line, $y = y_0 + (P - y_0)(1 - \exp(-kx))$, $y_0 = 0.85$, $P = 53.49$, $k = 0.018$). **f**, DNA nanotube bundle thickness ... exponential fit plotted as red line, $y = y_0 + (P - y_0)(1 - \exp(-kx))$, $y_0 = 11.79$, $P = 138.80$, $k = 0.017$.”

REVIEWERS' COMMENTS

Reviewer #1 (Remarks to the Author):

The authors have addressed all my concerns.

I recommend publication in Nature Communications.